# Governmental Taxation of Households Choosing between a National Currency and a Cryptocurrency

**Guizhou Wang and Kjell Hausken ***

Faculty of Science and Technology, University of Stavanger, 4036 Stavanger, Norway; pobewang@outlook.com
* Correspondence: kjell.hausken@uis.no; Tel.: +47-51-831632; Fax: +47-51-831550

**Abstract:** A game between a representative household and a government was analyzed. The household chose which fractions of two currencies to hold, e.g., a national currency such as a Central Bank Digital Currency (CBDC) and a global currency such as Bitcoin or Facebook's Diem, and chose the tax evasion probability for each currency. The government chose, for each currency, the probability of detecting and prosecuting tax evasion, the tax rate, and the penalty factor imposed on the household when tax evasion was successfully detected and prosecuted. The household's fraction of the national currency, the government's monitoring probability of the national currency, and the penalty factor imposed on the global currency, increased in the household's Cobb Douglas output elasticity for the national currency. The household's probabilities of tax evasion on both currencies increased in the government's Cobb Douglas output elasticity for the national currency. The government's taxation on both currencies decreased in the output elasticity for the national currency. High output elasticity for the national currency eventually induced the government to tax that currency more than the global currency. The household's probability of tax evasion on the global currency increased in the government's output elasticity for that currency. The household was less (more) likely to tax evade on the national (global) currency if the government valued taxation and penalty on the national (global) currency. The results are illustrated numerically where each of the eight parameter values was varied relative to a benchmark.

**Keywords:** digital currency; cryptocurrency; CBDC; Bitcoin; game theory; taxation; household; government

**JEL Classification:** C72; H26

## 1. Introduction

### 1.1. Background

Digital currencies are receiving increasing attention as central banks launch Central Bank Digital Currencies (CBDCs) (https://cbdctracker.org/, retrieved 7 April 2021), companies develop currencies (e.g., Facebook's Diem), and individuals, institutions, and others (e.g., Tesla, Grayscale, MicroStrategy, Square) buy Bitcoin and other cryptocurrencies. As of 7 April 2021, 9162 cryptocurrencies contributed to a market cap of $1.9 trillion (https://coinmarketcap.com/, retrieved 7 April 2021).

Cryptocurrencies work via the distributed ledger technology or blockchain. Blockchain is a decentralized technology spread across many nodes that manage and record transactions. The transactions are stored in multiple nodes that are permanent, verifiable, and unchangeable. Cryptocurrencies have no physical form, are typically not issued by a central authority, and are controlled through networks with varying degrees of decentralization. The first cryptocurrency was Bitcoin that emerged through the genesis block 3 January 2009 at 18:15:05 UTC.

Advantages of cryptocurrency included typical avoidance of inflation (e.g., through a fixed limited supply for Bitcoin or burning coins for the Binance coin), self-governance,

disintermediation (no central party), security, privacy, cost-effective transaction modes (especially for cross borders payments), instant or quick, and 24/7/365 accessibility, etc. Disadvantages of cryptocurrencies include possible use for illegal transactions (e.g., by applying privacy coins such as Monero), challenges of market fluctuations, no security or remedy in case of loss, limited scalability for some cryptocurrencies, etc.

Cryptocurrencies, and especially privacy coins like Monero, Verge, Zcash, etc., might enable tax evasion, which challenges regulators. Households might correctly or incorrectly assess and compare governments' abilities to monitor storage and transactions and enforce regulations for cryptocurrencies and government-issued currencies. Marian [1] suggests that cryptocurrencies could replace tax havens as the weapon-of-choice for tax-evaders.

These developments induce households to determine what fractions of each currency to hold, how to evade tax on each currency, and induce governments to determine how to tax, monitor tax evasion, and punish tax evasion, on each currency.

### 1.2. Contribution

This article models a game between a representative household and a government. The household chooses three strategies, i.e., the fractions to hold and the probabilities of tax evasion for two currencies. The government chooses six strategies, i.e., tax rates, tax monitoring, and punishments for tax evasion, for two currencies. The national currency offers the most common usage within a nation, e.g., purchasing and selling goods and services, paying taxes, and saving for retirement. The global currency generally offers opportunities beyond the national borders, e.g., user autonomy, discretion, peer-to-peer focus, and tax evasion.

The players' choices cause the household to assess four fractions for each currency; i.e., legally permitted for the household to keep, successful tax evasion, unsuccessful tax evasion, and the tax fraction paid voluntarily. The household has a Cobb Douglas expected utility with one output elasticity for each currency. The government has a Cobb Douglas expected utility with four output elasticities, i.e., one output elasticity for each currency reflecting its identification with the household, and one output elasticity for each currency reflecting its preference for taxation and penalties on unsuccessful tax evasion.

This article proposes a new way to formulate the government's utility. The government represents its households. Hence, the government is to some extent assumed to identify with each household, and benefits when the household benefits. The government also benefits from the household paying taxes, and benefits from the household paying a penalty when the government successfully monitors, and thus detects and prosecutes tax evasion.

The article analytically determines how eight parameters, intended to capture the phenomenon, impact the players' nine strategies and two expected utilities. Sensitivity analysis shows the variation in the government's monitoring probabilities, tax rates, penalty factors, and expected utility, and the household's fractions of the two currencies and the probability of tax evasion for each currency, as each parameter value varies relative to a benchmark. The results are discussed in terms of economic intuition and policy implications. The article contributes to all four areas of the literature reviewed in the next section.

### 1.3. Literature

The literature is divided into four areas, i.e., CBDC and cryptocurrencies, currency competition, game theory analyses, and governmental taxation.

#### 1.3.1. CBDC and Cryptocurrencies

This article relates to this literature by considering one national currency that can be interpreted to be a CBDC and one global currency that can be interpreted to be a cryptocurrency.

Blakstad and Allen [2] summarized the possibilities and risks offered by cryptocurrencies for central banks and individuals.

Brunnermeier and Niepelt [3] developed a generic framework of money, liquidity, seignories rents, and financial frictions. They provided sufficient conditions for the equivalence of monetary systems. They proposed that the introduction of CBDC could reduce run risk on banks, rather than increasing it.

Asimakopoulos et al. [4] developed a Dynamic Stochastic General Equilibrium (DSGE) model to assess the economic consequences of cryptocurrencies. Applying Bayesian techniques using US and crypto markets monthly data for the period 2013:M6-2019:M3, they found a strong substitution effect between the real balances of government currency and the real balances of cryptocurrency.

Sapkota and Grobys [5] divided cryptocurrencies into privacy and non-privacy coins. They explored whether asset market equilibria exist in the cryptocurrency markets. By analyzing ten cryptocurrencies with the highest market capitalization in each submarket in the 2016–2018 period, they found that privacy coins and non-privacy coins expressed two distinct unrelated market equilibria.

Allen et al. [6] enumerated the fundamental technical design challenges facing CBDC designers, with a particular focus on performance, privacy, and security. They summarized the main potential benefits of CBDC, namely, efficiency, a broader tax base, flexible monetary policy, payment backstop, and financial inclusion.

### 1.3.2. Currency Competition

This article relates to this literature by considering competition between one national currency and one global currency, in the sense that each household chooses optimally how much to hold of each.

Gandal and Halaburda [7] evaluated the impact of network effects on competition in the cryptocurrency market. They found no winner-take-all effects in the early period since November 2013 (when data collection started) until April 2014, but strong network effects and winner-take-all dynamics from April 2014 until February 2016.

Benigno [8] stated that multiple currencies could compromise the primary function of a central bank. Additionally, they found that with many competing currencies issued by profit-maximizing actors, both the nominal interest rate and the inflation could not be manipulated, but were instead determined by structural factors, such as the intertemporal discount factor, the exit rate, and the fixed entry cost.

Fernández-Villaverde and Sanches [9] considered competition between privately issued fiat currencies. They found that an equilibrium existed in which price stability was consistent with competing private monies, and also, that a continuum of equilibrium trajectories existed with the property, such that the value of private currencies monotonically converged to zero.

Benigno et al. [10] evaluated a two-country economy with complete markets, two national currencies, and a global cryptocurrency. They suggest that deviating from interest rate equality might imply approaching the zero lower bound or the abandonment of the national currency, referred to as Crypto-Enforced Monetary Policy Synchronization (CEMPS). Hence, the impossibility of jointly ensuring a fixed exchange rate, free capital flows, and an independent monetary policy (the classic Impossible Trinity) becomes even less reconcilable.

### 1.3.3. Game Theory Analyses

This article relates to this literature by considering a game between a government and a representative household.

Wang [11] set up a game theory model to analyze the implications of tax evasion for the optimal design of CBDC. He discussed several scenarios where CBDC had different anonymity compared to cash. For example, if CBDC offered less anonymity than cash, introducing CBDC would decrease tax evasion. If CBDC provided a high level of anonymity but low interest rate, then it would decrease the agents' output. However, if CBDC

offered low anonymity and a high interest rate, it would increase the output and aggregate the welfare.

Zhang et al. [12] assessed the tax preferences of enterprise income for comprehensive utilization of resources. They theoretically explored the game tax preference policy for energy conservation and emission reduction. They found that increasing camouflage cost and expected cost of risk could effectively prevent the generation of enterprise frauds.

Caginalp and Caginalp [13] determined the game theory equilibria for cryptocurrencies. The players divided their assets between the home currency and the cryptocurrency. The government decided the probability of seizing a fraction of the players' assets. The conditions for existence and uniqueness of Nash equilibria were established.

Wang and Hausken [14] analyzed competition between a national currency and a global currency, both of which had specific characteristics in an economy. The replicator equation was used to illustrate how conventionalists (which prefer to be in the majority) tend to compete against the pioneers and criminals (which prefer to be in the minority), under various conditions.

Welburn and Hausken [15,16] theoretically analyzed the economic crises game, assuming six kinds of players, i.e., countries, central banks, banks, firms, households, and financial inter-governmental organizations. Players have strategies such as setting interest rates, lending, borrowing, producing, consuming, investing, importing, exporting, defaulting, and penalizing default.

### 1.3.4. Taxation

This article related to this literature by considering how a government taxes, monitors, and punishes tax evasion, and how a representative household might evade tax on two currencies.

### Reviews

Alm [17] reviewed how to measure, explain, and control tax evasion. The examples were to analyze shadow economies, experimental methods, survey evidence, assess currency demand, and trace evasion in transactions financed by currencies.

Andreoni et al. [18] theoretically and empirically reviewed the literature on tax compliance. They pointed out that the theoretical models only served as rough guides for empirical research. They recommended more work on exploring the psychological, moral, and social impacts on tax compliance activities, more attention to the dynamic and complex institutional framework of tax compliance, and more empirical research outside the USA jurisdiction.

### Governmental Taxation

Brito et al. [19] analyzed the optimal income tax problem when consumers work for many periods. The results indicated that when the government commits to future tax schedules, intertemporal nonstationary tax schedules could relax the self-selection constraints and lead to Pareto improvements.

Lai and Liao [20] investigated the optimal capital income taxation in heterogeneous agent economies, featuring endogenous government spending. They pointed out that the long-run optimal capital tax rate should not be zero when the competitive equilibrium risk-free interest rate differed from the subjective time discount rate. The results could be extended to a wide range of model economies.

Liu [21] explored how government preferences affected the choices of capital tax rates in the presence of tax competition. The article suggests that countries emphasizing economic development tend to choose lower corporate income tax rates than countries emphasizing regional equality.

Raurich [22] developed an endogenous growth model with an endogenous labor supply. He pointed out that the dynamic equilibrium might exhibit local indeterminacy when labor income is heavily taxed.

Economides et al. [23] presented a general equilibrium model of endogenous growth with productive and non-productive public goods and services. They solved for Ramsey second-best optimal policy. The findings differed from the benchmark case of the social planner's first-best allocation and depended crucially on whether public goods and services were subject to congestion.

Chen and Guo [24] explored the theoretical interrelations between progressive income taxation and macroeconomic (in)stability. The results showed that progressive taxation operated like an automatic destabilizer that generated equilibrium indeterminacy and belief-driven fluctuations in the economy, which differed from traditional Keynesian-type stabilization policies.

Bacchetta and Perazzi [25] discussed a monetary reform in Switzerland. Based on a simple infinite-horizon open-economy model, they pointed out that a tradeoff existed between a reduction in distortionary labor taxes and an increase in the opportunity cost of holding money.

Tax Evasion and Punishment

Becker [26] and Hausken and Moxnes [27] recommended optimal public and private policies to combat illegal behavior. They showed that optimal enforcement depended on the cost of catching and convicting offenders, the nature of punishments, and the responses of offenders to changes in enforcement. Similarly, this article showed how households responded to punishments for tax evasion.

Allingham and Sandmo [28] explored static and dynamic aspects of the taxpayer's decisions on tax evasion. In the static model, they found that the penalty rate and the probability of detection were substitutes for each other. In the dynamic analysis, they showed that consistent rational individuals always declared more taxes than myopic short-sighted tax-evading individuals. Extending Allingham and Sandmo's [28] work, Yitzhaki [29] showed that if a penalty was imposed on the evaded tax, no contradiction existed between an income and a substitution effect. Furthermore, if the taxpayer had absolute risk aversion, which decreased with income, increased taxation causes decreased tax evasion. This article supported the finding, when varying how the government identified with the household's output elasticity for the national currency (see Section 4) and when varying the government's elasticity for the national currency, when valuing taxation and penalty on unsuccessful tax evasion (see Section 4), and otherwise supported the opposite result or that one variable did not vary when the other variable varied.

Myles and Naylor [30] set out a model of tax evasion that captured a benefit of conforming with non-evaders and of adhering to the social custom of non-evasion. They showed that both equilibria with no evasion and with taxpayers choosing to evade could exist. Similarly, this article showed how households might respond differently to the government's taxation, monitoring, and punishment.

Slemrod and Yitzhaki [31] presented theoretical models that integrate tax avoidance and evasion into the overall decision problem faced by taxpayers. They also developed a taxonomy of efficiency costs and introduced a general theory of optimal tax systems. They found that when the tax structure changed, individuals might change their consumption basket.

Experimental Work on Tax Evasion

Torgler [32] summarized experimental findings on tax morale and tax compliance, focusing on personal income tax morale, and social and institutional factors. He argued for the infeasibility of testing the predictions of the level of tax compliance models. In addition, social and institutional factors were important factors on tax compliance.

Kleven et al. [33] presented a tax enforcement field experiment in Denmark. They found that tax evasion was near zero for income subject to third-party reporting, and was much higher for self-reported income. In addition, marginal tax rates impacted tax evasion positively for self-reported income, but the effect was small compared to legal avoidance

and behavioral responses. Additionally, prior audits and threat-of-audit letters significantly impacted self-reported income, but did not impact third-party reported income.

Empirical Work on Tax Evasion

Ariyo and William [34] estimated that for 1975–2010, 42.54–79.32% of the Nigerian underground economy and tax evasion constituted 2.09–6.75% of the Gross Domestic Product.

Bittencourt et al. [35] found for 150 cases that less (more) financial development and a more (less) inflation caused a bigger (smaller) shadow economy with related tax evasion, during 1980–2009.

Hanlon et al. [36] assessed "round tripping" tax evasion where funds in offshore tax havens were invested in U.S. securities markets. They found that the incentives to evade U.S. taxation and expected costs of evasion detection affected the amount of foreign portfolio investment in U.S. debt and equity markets.

Tax Morale and Alternatives to Expected Utility Theory

Luttmer and Singhal [37] pointed out that apart from tax tools like tax rate, detection probality, and penalties imposed if evasion was detected, tax morale including nonpecuniary motivations were important factors in tax compliance decisions. Drawing on evidence from experiments, they demonstrated that tax morale operated through many underlying mechanisms.

Dhami and al-Nowaihi [38] contended that the expected utility theory failed to explain tax evasion activities. They found that the cumulative prospect theory provided a much more satisfactory explanation of tax evasion.

### *1.4. Article Organization*

Section 2 presents the model. Section 3 analyzes the model. Section 4 illustrates the solution. Section 5 discusses the results and provides economic intuition and policy implications. Section 6 concludes.

## 2. The Model

### *2.1. Two Currencies n and g*

Appendix A shows the nomenclature. Consider an economy with two available currencies. The first currency $n$ is national and offers the most common usage, and especially legal usage, within the economy. Examples of usage were for making various purchases or paying taxes. The government has complete control and dominance over the national currency $n$, e.g., by adjusting tax rates and inflation. We can think of the currency $n$ as a CBDC. The second currency $g$ is global and outside the control of the government. It offers more limited usage, e.g., cannot be used for all kinds of purchases, but offers other opportunities, e.g., user autonomy, discretion, peer-to-peer focus, no banking fees, tax evasion, black market payments, criminal activities, and a potentially high return. We might think of currency $g$ as a cryptocurrency such as Bitcoin, Zcash, or Facebook's Diem.

The household pays taxes for holding the two currencies, and can choose tax evasion with a probability for each currency. If tax evasion is detected and prosecuted by the government, the household has to pay a penalty. Owing to the features of the two currencies, the probabilities of tax evasion, tax rates, probabilities of detecting tax evasion, and penalty factors if tax evasion is detected, generally differ. Figure 1 illustrates the two currencies $n$ and $g$.

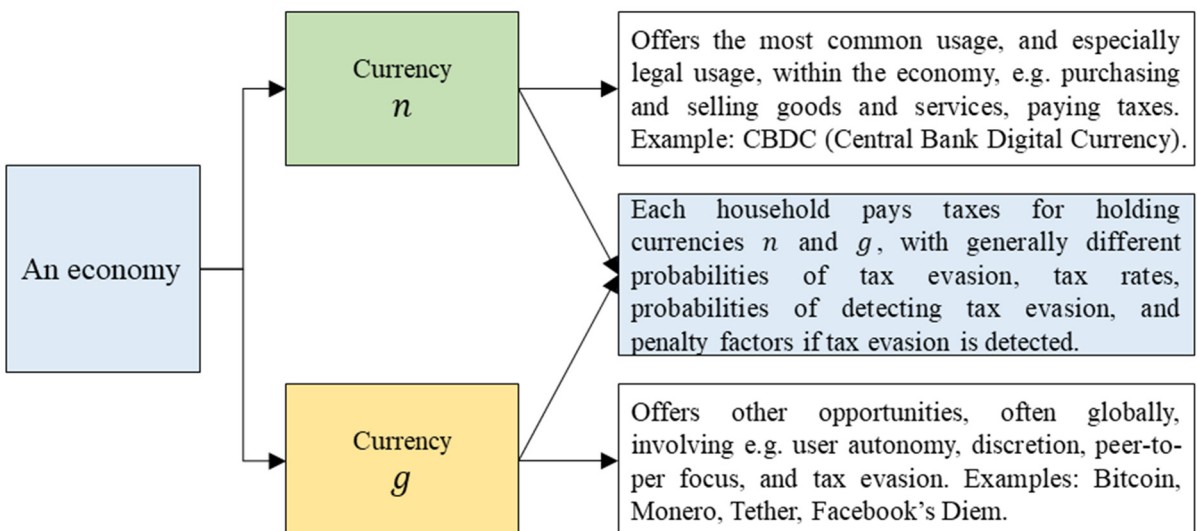

**Figure 1.** An economy with two currencies $n$ and $g$.

*2.2. Two Kinds of Players: Households and One Government*

Consider an economy with a representative household and a government. The household chooses the fraction to hold currency $n$, causing the remaining fraction to be held in currency $g$, and chooses the tax evasion probability for each currency. The government is the second player. It completely controls the national currency $n$, but has no control of the global currency $g$. However, the government can set the tax rates, the probabilities of detecting tax evasion, and the penalty factors if tax evasion is detected, for both currencies. We consider a non-cooperative one-period game. The households and government choose their strategies simultaneously and independently. The players are interlinked as in Figure 2.

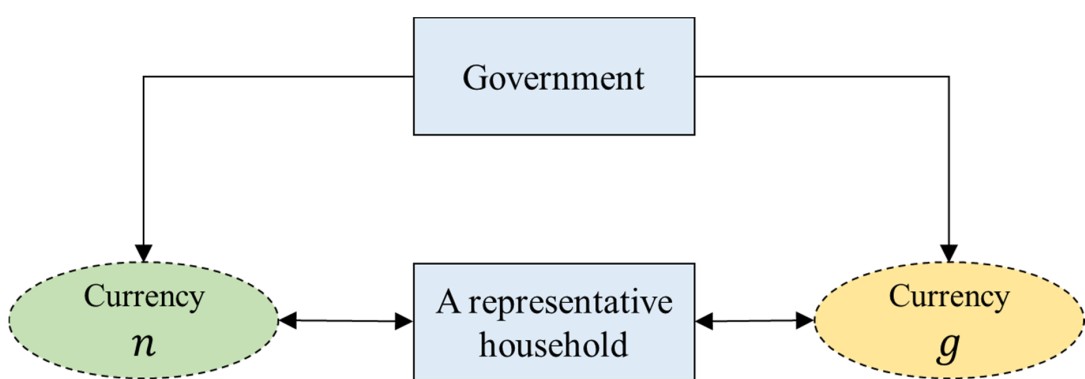

**Figure 2.** The government and a representative household involved in a national currency $n$ and a global currency $g$.

*2.3. The Players' Strategic Choices*

The representative household simultaneously chooses three strategies to maximize its expected utility $U$. It chooses its fraction $x$, $0 \leq x \leq 1$ of currency $n$, causing the remaining fraction $1 - x$ to be held in currency $g$. Additionally, it chooses the tax evasion probability $p_j$, $0 \leq p_j \leq 1$, for currency $j$, $j = n, g$.

The government chooses six strategies simultaneously to maximize its expected utility $u$. It chooses the probability $m_j$, $0 \leq m_j \leq 1$ of detecting and prosecuting tax evasion on currency $j$. Additionally, it chooses the tax rate $\tau_j$, $\tau_j \geq 0$ for currency $j$. Finally, it chooses the penalty factor $P_j$, $P_j \geq 0$, imposed on each household when tax evasion is successfully detected and prosecuted on currency $j$, $j = n, g$. Table 1 shows the players' strategies descriptions and strategy sets.

| Player | Strategies Description | Strategy Set |
|---|---|---|
| A representative household | Chooses its fraction $x$, $0 < x \leq 1$, of currency $n$, causing the remaining fraction $1 - x$, to be held in currency $g$.<br>Chooses the tax evasion probability $p_n$ for currency $n$ and tax evasion probability $p_g$ for currency $g$. | $\{x, p_n, p_g\}$ |
| Government | Chooses the probability $m_j$, $0 \leq m_j \leq 1$, of detecting and prosecuting tax evasion on currency $j$.<br>Chooses the tax rate $\tau_j$, $\tau_j \geq 0$, for currency $j$.<br>Chooses the penalty factor $P_j$, $P_j \geq 0$, imposed on each household when tax evasion is successfully detected and prosecuted on currency $j$, $j = n, g$. | $\left\{ \begin{matrix} m_n, \ m_g, \tau_n, \\ \tau_g, P_n, P_g \end{matrix} \right\}$ |

### 2.4. The Household's Strategies and Expected Utility

Assume that a representative household evades taxes on currency $j$ with probability $p_j$, $0 \leq p_j \leq 1$, $j = n, g$, which is detected and prosecuted by the government with probability $m_j$, $0 \leq m_j \leq 1$. With a tax rate $\tau_j$, $0 \leq \tau_j \leq 1$, for currency $j$, the household's expected tax payment fraction on currency $j$ is $(1 - p_j)\tau_j$, paid voluntarily. With zero government detection $m_j = 0$, the household's expected income fraction from tax evasion on currency $j$ is $p_j\tau_j$. With 100% government detection and prosecution $m_j = 1$, the household's expected income fraction from tax evasion on currency $j$ is 0. Generally, the household's expected income fraction from tax evasion on currency $j$ is $(1 - m_j)p_j\tau_j$, i.e., successful tax evasion. Hence, the household's expected expense fraction without penalty from unsuccessful tax evasion on currency $j$ is $m_jp_j\tau_j$. We assume that the government penalizes unsuccessful tax evasion by adjusting $m_jp_j\tau_j$ in two ways. First, $m_jp_j\tau_j$ is multiplied with a penalty factor $P_j$, $P_j \geq 0$, chosen by the government as a free choice variable. Second, $P_jm_jp_j\tau_j$ is assumed to depend on the representative household's tax evasion probability $p_j$ in a more flexible manner by replacing $p_j$ with $p_j^{\lambda_j}$, where $p_j$ is a parameter, which gives $m_j\tau_jP_jp_j^{\lambda_j}$ as the household's expense from unsuccessful tax evasion. We require $\lambda_j \geq 0$ since the household's expected expense for tax evasion should increase as the household's tax evasion probability increases, $\partial\left(m_j\tau_jP_jp_j^{\lambda_j}\right)/p_j \geq 0$. Tax evasion should not be beneficial. We might interpret $P_jp_j^{\lambda_j-1}$ as the government's penalty, which is multiplied with the household's expected expense fraction $m_jp_j\tau_j$ from unsuccessful tax evasion on currency $j$, to give $m_j\tau_jP_jp_j^{\lambda_j}$. Hence, the household keeps a fraction

$$f_j = 1 - (1 - p_j)\tau_j - m_j\tau_jP_jp_j^{\lambda_j} \tag{1}$$

of currency $j$, which is multiplied with the fraction $x$ of currency $n$, and multiplied with the fraction $1 - x$ of currency $g$, to determine how much of the two currencies $n$ and $g$ the household owns. The fraction $f_j$ is positive when $P_j \leq \frac{1-(1-p_j)\tau_j}{m_j\tau_jp_j^{\lambda_j}}$, and is otherwise negative.

We apply the Cobb Douglas expected utility for both players, since it is widely used within economics and since it explicitly captures tradeoff players strike between multiple conflicting or partly conflicting objectives. For the household that includes which currencies to hold and with which probabilities to tax evade, assume that the household has a Cobb Douglas expected utility with output elasticity $\alpha$, $0 \leq \alpha \leq 1$, associated with currency $n$, and $1 - \alpha$ associated with currency $g$, i.e.,

$$U = \begin{cases} \left( \left( 1 - (1-p_n)\tau_n - m_n\tau_n P_n p_n^{\lambda_n} \right) x \right)^{\alpha} \\ \times \left( \left( 1 - (1-p_g)\tau_g - m_g\tau_g P_g p_g^{\lambda_g} \right)(1-x) \right)^{1-\alpha} \\ \quad if \ P_j \le \frac{1-(1-p_j)\tau_j}{m_j\tau_j p_j^{\lambda_j+1}}, j = n, g \\ 0 \ otherwise \end{cases}$$
(2)

where $U = 0$ means that the penalty factor $P_j$ is so high that the household goes into debt. This is illustrated in Figure 3. The household's three free choice variables are its fraction $x$ of currency $n$, which causes the remaining fraction $1 - x$ to be held in currency $g$, and its tax evasion probability $p_j$ for currency $j$, $j = n, g$.

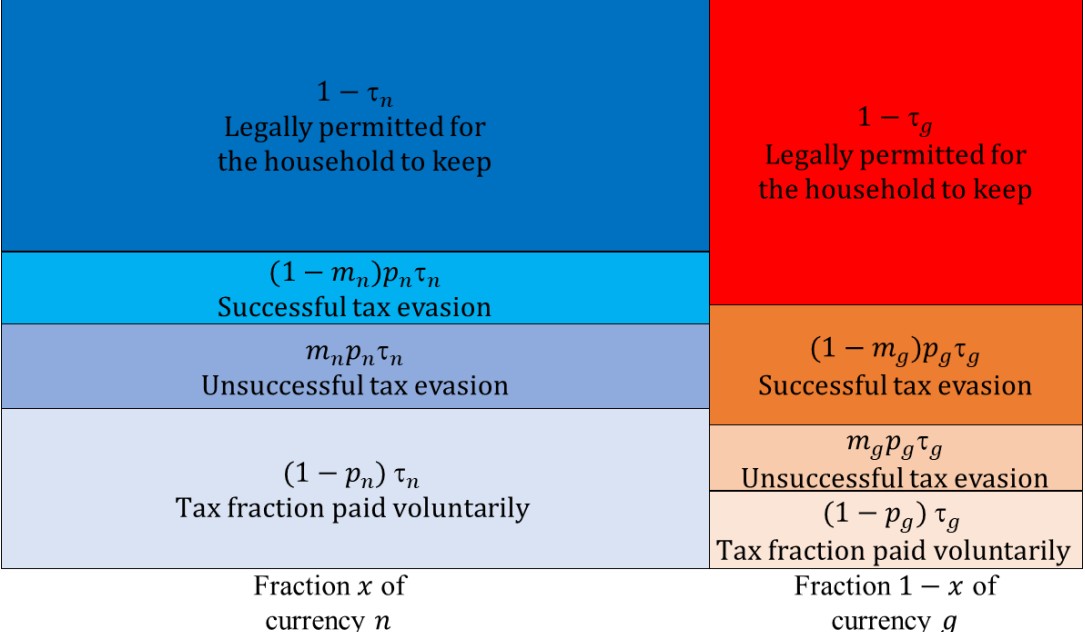

**Figure 3.** Fractions $x$ and $1 - x$ of the household's currencies $n$ and $g$, each divided into four subgroup fractions, i.e., $1 - \tau_j$ as legally permitted for the household to keep, $\left(1 - m_j\right)p_j\tau_j$ as successful tax evasion, $m_j p_j\tau_j$ as unsuccessful tax evasion, and $\left(1 - p_j\right)\tau_j$ as the tax fraction paid voluntarily, $j = n, g$.

The output elasticities $\alpha$ and $1 - \alpha$ for the two currencies $n$ and $g$ account in a deep sense for the benefits and costs of holding, acquiring, and transacting with the two currencies. Cryptocurrencies are freely available. Once acquired, no costs exist of holding them, and interest might be earned. If we think of currency $g$ as Bitcoin, these benefits and costs changed since the genesis block in 2009. The early Bitcoin adopters operated in a segmented market, possessing competence beyond the majority of households. Over the last years, the market has broadened, become less segmented, is more easily accessible through multiple entry points, and is more user-friendly. Users learned to use crypto wallets, which are of five types—mobile, desktop, paper, hardware, online, and mobile wallets. Users operate on platforms and exchanges such as ImToken, Metamask, TrustWallet, TokenPlus, Binance, OKEx, Huobi, Coinbase, etc. Users download apps such as Abra from the internet on their cellphone, and create their own cryptocurrency addresses, where they buy, sell, exchange, and earn interest on cryptocurrencies. Buying cryptocurrencies has become similar to buying stocks and is almost costless. Cryptocurrencies are gradually incorporated into the conventional financial system, exemplified with Paypal, which currently offers Bitcoin, Bitcoin Cash, Ethereum, and Litecoin. To the extent the representative household perceives holding a global currency $g$ such as Bitcoin as less straightforward than holding a

government-issued national currency $n$, the household assigns lower output elasticity $1 - \alpha$ to the global currency $g$, and thus higher output elasticity $\alpha$ to the national currency $n$.

### 2.5. The Government's Strategies and Expected Utility

The challenge in modeling the government is that it cannot identify 100% with each household individually, because of the collective action dilemma, including the objective of maximizing the expected utility or welfare of all households. The government also cannot minimize the expected utility of each household since then it will not be reelected. Hence, we assume that the government to some extent identifies with and represents each household, and benefits when the household benefits. A straightforward way of accomplishing that objective is to incorporate the household's expected utility $U$ in Equation (2) into the government's expected utility $u$. That implicitly means that the government to some extent, as determined by the parameters and the players' strategic choices, internalizes all advantages of the household, including the advantage of evading taxes for the household. Since internalizing that advantage cannot be taken too far, we assume that the government also benefits from the household paying taxes, and benefits from the household paying a penalty when the government successfully monitors, and thus detects and prosecutes tax evasion. The government finally has a cost expenditure of choosing the monitoring probability $m_j$, $j = n, g$. These multiple conflicting or partly conflicting objectives of the government are obtained by assuming a more extensive Cobb Douglas expected utility for the government, expressed per household as

$$
U = \begin{cases}
\left( \left( 1 - (1 - p_n)\tau_n - m_n \tau_n P_n p_n^{\lambda_n} \right) x \right)^{\beta_n} \\
\times \left( \left( 1 - (1 - p_g)\tau_g - m_g \tau_g P_g p_g^{\lambda_g} \right)(1 - x) \right)^{\beta_g} \\
\times \left( \left( (1 - p_n)\tau_n + m_n \tau_n P_n p_n^{\lambda_n} \right) x - a_n m_n \right)^{\gamma_n} \\
\times \left( \left( (1 - p_g)\tau_g + m_g \tau_g P_g p_g^{\lambda_g} \right)(1 - x) - a_g m_g \right)^{1 - \beta_n - \beta_g - \gamma_n} \\
\qquad \text{if } P_j \le \frac{1 - (1 - p_j)\tau_j}{m_j \tau_j p_j^{\lambda_j}}, j = n, g \\
0 \text{ otherwise}
\end{cases}
\tag{3}
$$

which has four multiplicative terms. The first two terms in Equation (3) are equivalent to the two terms in Equation (2), except that $\alpha$ and $1 - \alpha$ are replaced with $\beta_n$ and $\beta_g$, respectively, $0 \le \beta_n, \beta_g \le 1$. That replacement means that although the government identifies with the household, the government is enabled to prioritize differently and have other output elasticities for the two currencies $n$ and $g$ than the household. For the special case when the government has the same ratio $\alpha/(1 - \alpha) = \beta_n/\beta_g$ between the two currencies $n$ and $g$ as the household, we get

$$
\frac{\alpha}{1 - \alpha} = \frac{\beta_n}{\beta_g} \Leftrightarrow \alpha = \frac{\beta_n}{\beta_n + \beta_g}
\tag{4}
$$

which we do not require the government to adhere to. The third and fourth terms in Equation (3), for currencies $n$ and $g$, respectively, express that the government maximizes the sum of two terms and a subtracted third term raised to the output elasticities $\gamma_n$ and $1 - \beta_n - \beta_g - \gamma_n$, respectively, $0 \le \gamma_n \le 1, 0 \le 1 - \beta_n - \beta_g - \gamma_n \le 1$, for currencies $n$ and $g$. Term 1 is the household's tax fraction paid voluntarily, multiplied with the currency fraction, i.e., $(1 - p_n)\tau_n x$ and $(1 - p_g)\tau_g(1 - x)$, for currencies $n$ and $g$, respectively. Term 2 is the household's unsuccessful tax evasion multiplied with the penalty and currency fraction, i.e., $m_n \tau_n P_n p_n^{\lambda_n} x$ and $m_g \tau_g P_g p_g^{\lambda_g}(1 - x)$, for currencies $n$ and $g$, respectively. Term 3 is the household's unit cost $a_j$, $a_j \ge 0$, of choosing the monitoring probability $m_j$, multiplied with $m_j$, $j = n, g$. Since $m_j$ is a probability, the unit cost $a_j$ has to be scaled so that $0 \le m_j \le 1$.

The government's six free choice variables are its probability $m_j$ of detecting and prosecuting tax evasion on currency $j$, the tax rate $\tau_j$ on currency $j$, and the penalty factor $P_j$ imposed on each household when tax evasion is successfully detected and prosecuted on currency $j$, $j = n, g$. The government and each household choose their free choice variables simultaneously and independently. Analyzing such a stationary situation reflects reality in the sense that governments in general, and households over time, adapt their preferences and strategies to each other, making it difficult to state that one player chooses a strategy over some other player.

## 3. Analyzing the Model

### 3.1. Analyzing the Household

Appendix B shows that the household chooses to hold the fraction

$$
x = \begin{cases} \alpha \ if \ P_j \le \frac{1-(1-p_j)\tau_j}{m_j \tau_j p_j^{\lambda_j}}, j = n, g \\ undetermined \ otherwise \end{cases}
\tag{5}
$$

of currency $n$, and thus the remaining fraction $1 - x$ of currency $g$, and chooses the probability

$$
p_j = \begin{cases} \frac{1}{\left(m_j P_j \lambda_j\right)^{1/(\lambda_j - 1)}} \ if \ P_j \le \frac{1-(1-p_j)\tau_j}{m_j \tau_j p_j^{\lambda_j}} \ and \ 0 \le p_j \le 1, j = n, g \\ undetermined \ or \ 1 \ otherwise \end{cases}
\tag{6}
$$

of tax evasion on currency $j$, $j = n, g$.

### 3.2. Analyzing the Government

Appendix C shows that the government chooses the free choice variables

$$
\begin{aligned}
&m_n = \frac{x}{a_n}, m_g = \frac{1-x}{a_g}, \tau_n = \frac{\gamma_n}{(1-p_n)(\beta_n+\gamma_n)}, \tau_g = \frac{1-\beta_n-\beta_g-\gamma_n}{(1-p_g)(1-\beta_n-\gamma_n)}, \\
&P_n = \frac{a_n(1-p_n)\beta_n}{p_n^{\lambda_n} x \gamma_n}, P_g = \frac{a_g(1-p_g)\beta_g}{p_g^{\lambda_g}(1-x)(1-\beta_n-\beta_g-\gamma_n)}, \\
&0 \le m_n \le 1 \Leftrightarrow a_n \ge x, \ 0 \le m_g \le 1 \Leftrightarrow a_g \ge 1 - x, \ 0 \le \tau_n \le 1 \Leftrightarrow 0 \le p_n \le \frac{\beta_n}{\beta_n+\gamma_n}, \\
&0 \le \tau_g \le 1 \Leftrightarrow 0 \le p_g \le \frac{\beta_g}{1-\beta_n-\gamma_n}
\end{aligned}
\tag{7}
$$

### 3.3. Analyzing the Household and Government Together

**Property 1.** *The household's and the government's strategies are*

$$
\begin{aligned}
&x = \alpha, p_n = \frac{\lambda_n \beta_n}{\lambda_n \beta_n + \gamma_n}, p_g = \frac{\lambda_g \beta_g}{1-\beta_n-\gamma_n-(1-\lambda_g)\beta_g}, \\
&m_n = \frac{\alpha}{a_n}, m_g = \frac{1-\alpha}{a_g}, \tau_n = \frac{\lambda_n \beta_n + \gamma_n}{\beta_n + \gamma_n}, \tau_g = \frac{1-\beta_n-\gamma_n-(1-\lambda_g)\beta_g}{1-\beta_n-\gamma_n}, \\
&P_n = \frac{a_n}{\lambda_n \alpha}\left(\frac{\lambda_n \beta_n}{\lambda_n \beta_n + \gamma_n}\right)^{1-\lambda_n}, P_g = \frac{a_g}{\lambda_g(1-\alpha)}\left(\frac{\lambda_g \beta_g}{1-\beta_n-\gamma_n-(1-\lambda_g)\beta_g}\right)^{1-\lambda_g}, \\
&U = u = 0, a_n \ge \alpha, \ a_g \ge 1 - \alpha, \ 0 \le \lambda_j \le 1, \ j = n, g
\end{aligned}
\tag{8}
$$

**Proof.** Appendix D. □

**Property 2.** (1): $\frac{\partial x}{\partial \alpha} \ge 0$, $\frac{\partial(1-x)}{\partial \alpha} \le 0$, $\frac{\partial m_n}{\partial \alpha} \ge 0$, $\frac{\partial m_g}{\partial \alpha} \le 0$, $\frac{\partial P_n}{\partial \alpha} \le 0$, $\frac{\partial^2 P_n}{\partial \alpha^2} \ge 0$, $\frac{\partial P_g}{\partial \alpha} \ge 0$, $\frac{\partial^2 P_g}{\partial \alpha^2} \ge 0$, $\frac{\partial p_n}{\partial \alpha} = \frac{\partial p_g}{\partial \alpha} = \frac{\partial \tau_n}{\partial \alpha} = \frac{\partial \tau_g}{\partial \alpha} = 0$. (2): $\frac{\partial p_n}{\partial \lambda_n} \ge 0$, $\frac{\partial^2 p_n}{\partial \lambda_n^2} \ge 0$, $\frac{\partial \tau_n}{\partial \lambda_n} \ge 0$, $\frac{\partial x}{\partial \lambda_n} = \frac{\partial(1-x)}{\partial \lambda_n} = \frac{\partial p_g}{\partial \lambda_n} = \frac{\partial m_n}{\partial \lambda_n} = \frac{\partial m_g}{\partial \lambda_n} = \frac{\partial \tau_g}{\partial \lambda_n} = \frac{\partial P_g}{\partial \lambda_n} = 0$. (3): $\frac{\partial p_g}{\partial \lambda_g} \ge 0$, $\frac{\partial^2 p_g}{\partial \lambda_g^2} \ge 0$, $\frac{\partial \tau_g}{\partial \lambda_g} \ge 0$, $\frac{\partial x}{\partial \lambda_g} = \frac{\partial(1-x)}{\partial \lambda_g} = \frac{\partial p_n}{\partial \lambda_g} = \frac{\partial m_n}{\partial \lambda_g} = \frac{\partial m_g}{\partial \lambda_g} = \frac{\partial \tau_n}{\partial \lambda_g} = \frac{\partial P_n}{\partial \lambda_g} = 0$. (4): $\frac{\partial p_n}{\partial \beta_n} \ge 0$, $\frac{\partial^2 p_n}{\partial \beta_n^2} \le 0$, $\frac{\partial p_g}{\partial \beta_n} \ge 0$, $\frac{\partial^2 p_g}{\partial \beta_n^2} \ge 0$, $\frac{\partial \tau_n}{\partial \beta_n} \le 0$, $\frac{\partial^2 \tau_n}{\partial \beta_n^2} \ge 0$, $\frac{\partial \tau_g}{\partial \beta_n} \le 0$, $\frac{\partial^2 \tau_g}{\partial \beta_n^2} \le 0$, $\frac{\partial P_n}{\partial \beta_n} \ge 0$, $\frac{\partial^2 P_n}{\partial \beta_n^2} \le 0$, $\frac{\partial P_g}{\partial \beta_n} \ge 0$, $\frac{\partial x}{\partial \beta_n} = \frac{\partial(1-x)}{\partial \beta_n} = \frac{\partial m_n}{\partial \beta_n} = \frac{\partial m_g}{\partial \beta_n} = 0$. (5): $\frac{\partial p_g}{\partial \beta_g} \ge 0$, $\frac{\partial^2 p_g}{\partial \beta_g^2} \ge$

$$0, \frac{\partial \tau_g}{\partial \beta_g} \leq 0, \frac{\partial P_g}{\partial \beta_g} \geq 0, \frac{\partial x}{\partial \beta_g} = \frac{\partial (1-x)}{\partial \beta_g} = \frac{\partial p_n}{\partial \beta_g} = \frac{\partial m_n}{\partial \beta_g} = \frac{\partial m_g}{\partial \beta_g} = \frac{\partial \tau_n}{\partial \beta_g} = \frac{\partial P_n}{\partial \beta_g} = 0. \quad (6)$$

$$\frac{\partial p_n}{\partial \gamma_n} \leq 0, \frac{\partial^2 p_n}{\partial \gamma_n^2} \geq 0, \frac{\partial p_g}{\partial \gamma_n} \geq 0, \frac{\partial^2 p_g}{\partial \gamma_n^2} \geq 0, \frac{\partial \tau_n}{\partial \gamma_n} \geq 0, \frac{\partial^2 \tau_n}{\partial \gamma_n^2} \leq 0, \frac{\partial \tau_g}{\partial \gamma_n} \leq 0, \frac{\partial^2 \tau_g}{\partial \gamma_n^2} \leq 0, \frac{\partial P_n}{\partial \gamma_n} \leq 0, \frac{\partial^2 P_n}{\partial \gamma_n^2} \geq 0,$$

$$\frac{\partial P_g}{\partial \gamma_n} \geq 0, \frac{\partial^2 P_g}{\partial \gamma_n^2} \geq 0, \frac{\partial x}{\partial \gamma_n} = \frac{\partial (1-x)}{\partial \gamma_n} = \frac{\partial m_n}{\partial \gamma_n} = \frac{\partial m_g}{\partial \gamma_n} = 0. \quad (7): \frac{\partial m_n}{\partial a_n} \leq 0, \frac{\partial^2 m_n}{\partial a_n^2} \geq 0, \frac{\partial P_n}{\partial a_n} \geq 0, \frac{\partial x}{\partial a_n} =$$

$$\frac{\partial (1-x)}{\partial a_n} = \frac{\partial p_n}{\partial a_n} = \frac{\partial p_g}{\partial a_n} = \frac{\partial m_g}{\partial a_n} = \frac{\partial \tau_n}{\partial a_n} = \frac{\partial \tau_g}{\partial a_n} = \frac{\partial P_g}{\partial a_n} = 0. \quad (8): \frac{\partial m_g}{\partial a_g} \leq 0, \frac{\partial^2 m_g}{\partial a_g^2} \geq 0, \frac{\partial P_g}{\partial a_g} \geq 0, \frac{\partial x}{\partial a_g} =$$

$$\frac{\partial (1-x)}{\partial a_g} = \frac{\partial p_n}{\partial a_g} = \frac{\partial p_g}{\partial a_g} = \frac{\partial m_n}{\partial a_g} = \frac{\partial \tau_n}{\partial a_g} = \frac{\partial \tau_g}{\partial a_g} = \frac{\partial P_n}{\partial a_g} = 0.$$

**Proof.** Follows from Equations (A12)–(A19) in Appendix E. □

Property 2 states that, first, the household's fraction $x$ of currency $n$, the government's monitoring probability $m_n$ of currency $n$, and the government's penalty factor $P_g$ imposed on each household's holding of currency $g$, increase linearly, linearly, and convexly in the household's output elasticity $\alpha$ for currency $n$. Conversely, the household's fraction $1 - x$ of currency $g$, the government's monitoring probability $m_g$ of currency $g$, and the government's penalty factor $P_n$ imposed on each household's holding of currency $n$, decrease linearly, linearly, and convexly in $\alpha$. The remaining variables are independent of $\alpha$.

Second and third, the household's probability $p_j$ of tax evasion on currency $j$ and the government's taxation $\tau_j$ on currency $j$ increase concavely and linearly, respectively, in the exponential tax evasion parameter $\lambda_j$. The remaining variables except $P_j$ are independent of $\lambda_j$, $j = n, g$.

Fourth, the household's probabilities $p_n$ and $p_g$ of tax evasion on currencies $n$ and $g$ increase linearly and convexly in the government's output elasticity $\beta_n$ for currency $n$. The government's taxation $\tau_n$ and $\tau_g$ on currencies $n$ and $g$ decrease concavely and convexly in $\beta_n$. This decrease follows since increasing $\beta_n$ causes the government to identify more strongly with the household in Equation (3), and the household prefers low taxation. That the decrease is concave versus convex follows since high output elasticity $\beta_n$ for currency $n$ eventually induces the government to tax currency $n$ more than currency $g$. Furthermore, higher $\beta_n$ means lower output elasticity $1 - \beta_n - \beta_g - \gamma_n$ for the fourth term in Equation (3), which expresses lower government weight assigned to income from taxation and penalty on tax evasion associated with currency $g$. The government's penalty factors $P_n$ and $P_g$ imposed on each household's holding of currencies $n$ and $g$ increase concavely and convexly in $\beta_n$. The remaining variables are independent of $\beta_n$.

Fifth, the household's probability $p_g$ of tax evasion on currency $g$ increases convexly in the government's output elasticity $\beta_g$ for the same currency $g$, as currency $g$ becomes more valuable for the household. The government's taxation $\tau_g$ on currency $g$ decreases linearly in $\beta_g$, as the government identifies more strongly with the household and thus prefers to impose fewer costs on the household. The government's penalty factor $P_g$ imposed on each household's holding of currency $g$ increases convexly in $\beta_g$, as the government seeks to curtail the household's probability $p_g$ of tax evasion on currency $g$. The remaining variables are independent of $\beta_g$.

Sixth, the household's probabilities $p_n$ and $p_g$ of tax evasion on currencies $n$ and $g$ decreases concavely and increases convexly in the government's output elasticity $\gamma_n$ for currency $n$ when valuing taxation $\tau_n$ and valuing penalty $P_n$ on unsuccessful tax evasion on currency $n$. Thus, the household is less (more) likely to evade tax on currency $n$ ($g$) if the government values taxation $\tau_n$ ($\tau_g$) and penalty $P_n$ ($P_g$). The government's taxation $\tau_n$ and $\tau_g$ on currencies $n$ and $g$ increases concavely and decreases convexly in $\gamma_n$. The increase follows since increasing $\gamma_n$ causes the government to identify less strongly with the household's preference for low taxation $\tau_n$ on currency $n$, and instead to value taxation $\tau_n$ and penalty $P_n$. The decrease follows, conversely, since the government's higher valuation of taxation $\tau_n$ and penalty $P_n$ on currency $n$ implies a lower valuation of taxation $\tau_g$ and penalty $P_g$ on currency $g$. The government's penalty factors $P_n$ and $P_g$ imposed on each household's holding of currencies $n$ and $g$ which decreases concavely and increases convexly in $\gamma_n$. The remaining variables are independent of $\gamma_n$.

Seventh and eighth, the government's monitoring probability $m_j$ of currency $j$ decreases concavely in the unit cost $a_j$ of choosing $m_j$, while the government's penalty factor $P_j$ imposed on each household's holding of currency $j$ increases linearly. The remaining variables are independent of $a_j$. The remaining variables are independent of $a_j$, $j = n, g$.

## 4. Illustrating the Solution

To illustrate the solution in Property 1 in Section 3.3, this section alters the eight parameter values $\alpha, \lambda_n, \lambda_g, \beta_n, \gamma_n, \beta_g, a_n, a_g$ relative to the benchmark parameter values $\alpha = 4/5$, $\lambda_n = \lambda_g = 1/5$, $\beta_n = \gamma_n = 2/5$, $\beta_g = 1/10$, $a_n = a_g = 1$.

First, $\alpha = 4/5$ reflects that the national currency $n$ might be more common than the global currency $g$, in this illustration, four times more common. Second and third, $\lambda_n = \lambda_g = 1/5$ express that the household's expense $m_j \tau_j P_j p_j^{\lambda_j}$ from unsuccessful tax evasion increases concavely in the representative household's tax evasion probability $p_j$. Fourth, fifth, and sixth, $\beta_n = \gamma_n = 2/5 = \alpha/2$ and $\beta_g = (1-\alpha)/2 = 1/10$ preserve the same ratio $\alpha/(1-\alpha) = \beta_n/\beta_g = \gamma_n/(1-\beta_n-\beta_g-\gamma_n) = 4$ for how the household and government assign output elasticities to the national currency $n$ versus the global currency $g$. That is, both the household and the government assign a four times higher output elasticity to currency $n$ than to currency $g$ in their Cobb Douglas expected utilities $U$ and $u$, and the government does so for both first terms in Equation (3) pertaining to its identification with the household, and for the last two terms in Equation (3) pertaining to how the government benefits from taxation income and income from the household's penalty payment from unsuccessful tax evasion. Seventh and eighth, the government's unit effort costs $a_n = a_g = 1$ of choosing the monitoring probability $m_j$ are the simplest possible benchmarks that satisfy $a_n \geq \alpha$ and $a_g \geq 1 - \alpha$. In Figure 4, each of the eight parameter values is altered from its benchmark, while the other seven parameter values are kept at their benchmarks. Division of $P_j$ with 20 is for scaling purposes.

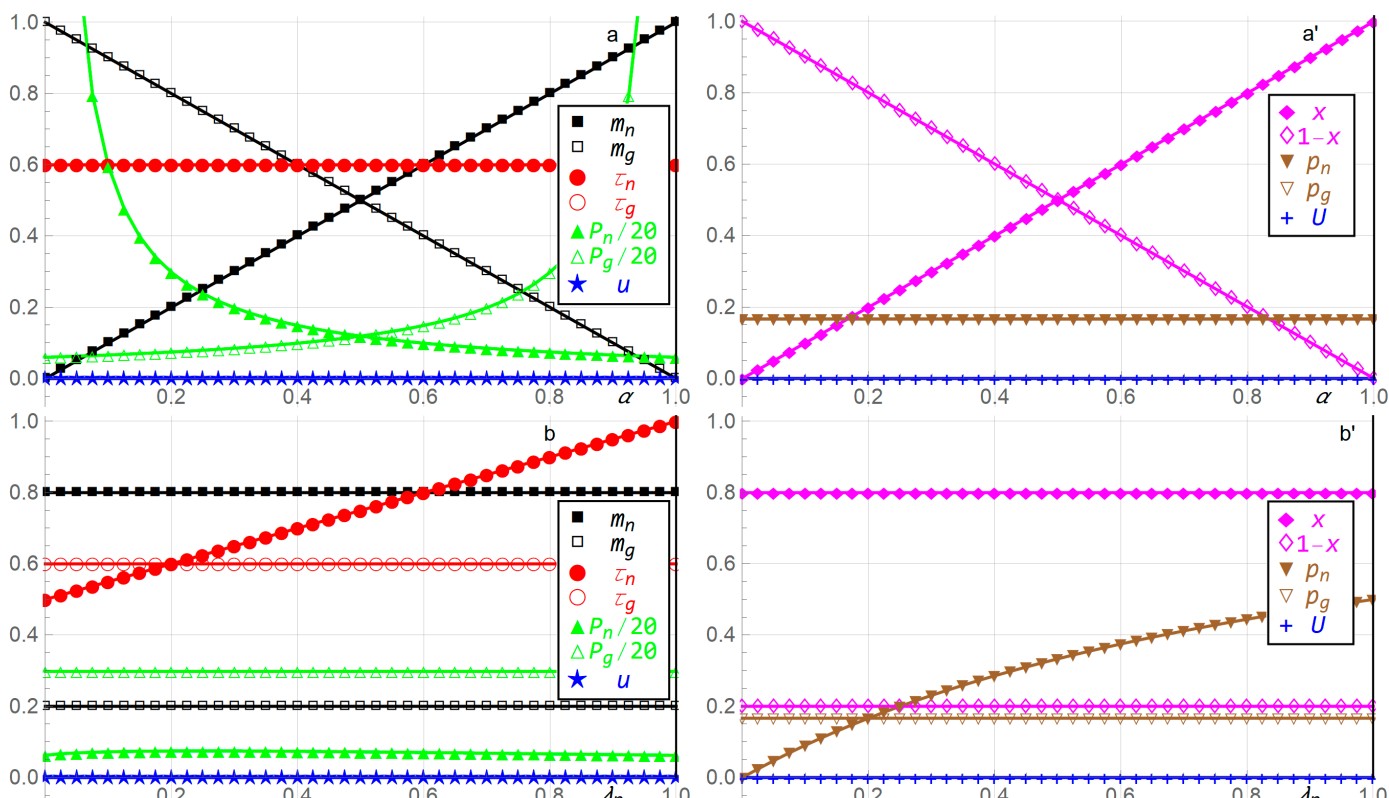

**Figure 4.** *Cont.*

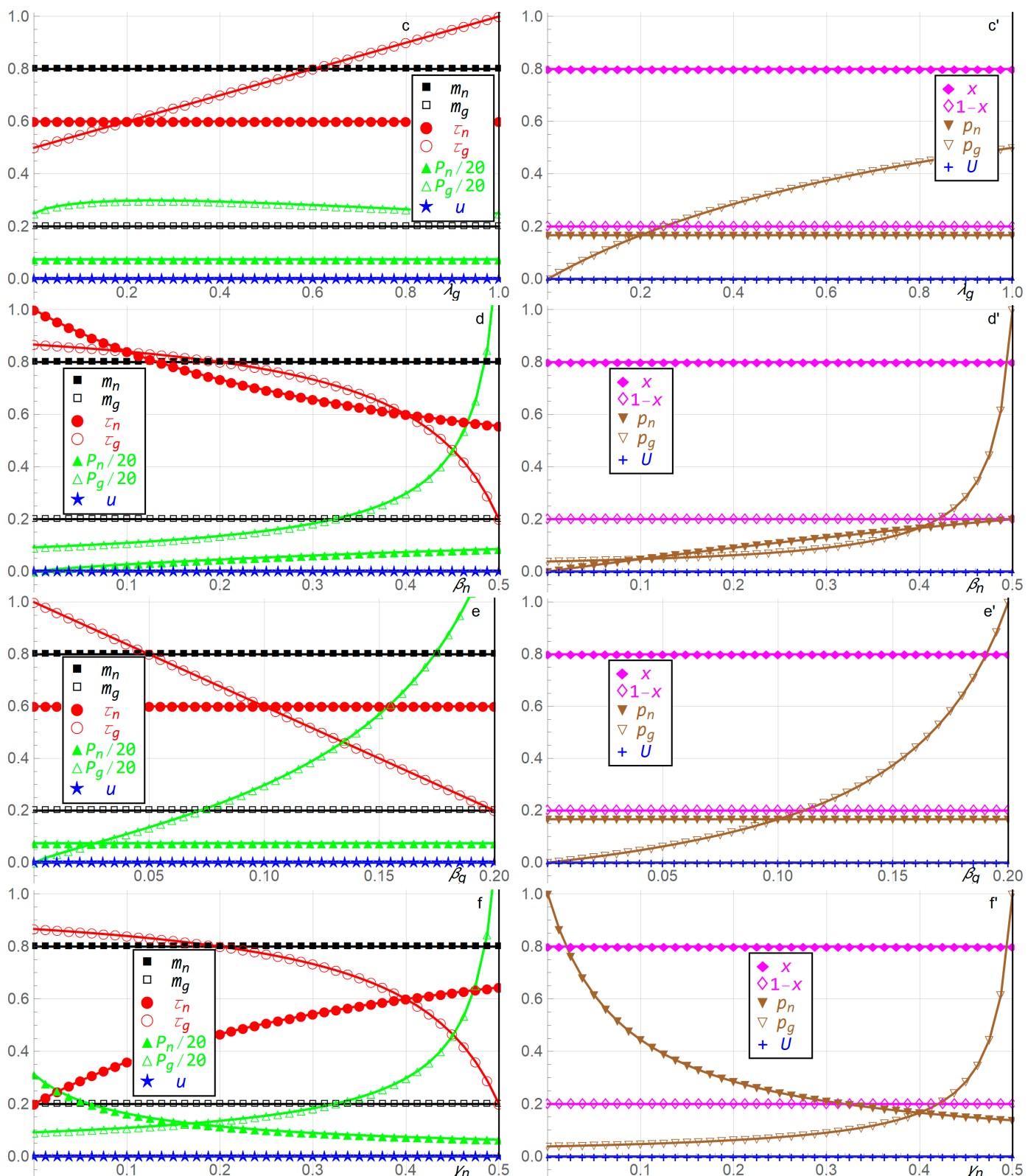

**Figure 4.** *Cont.*

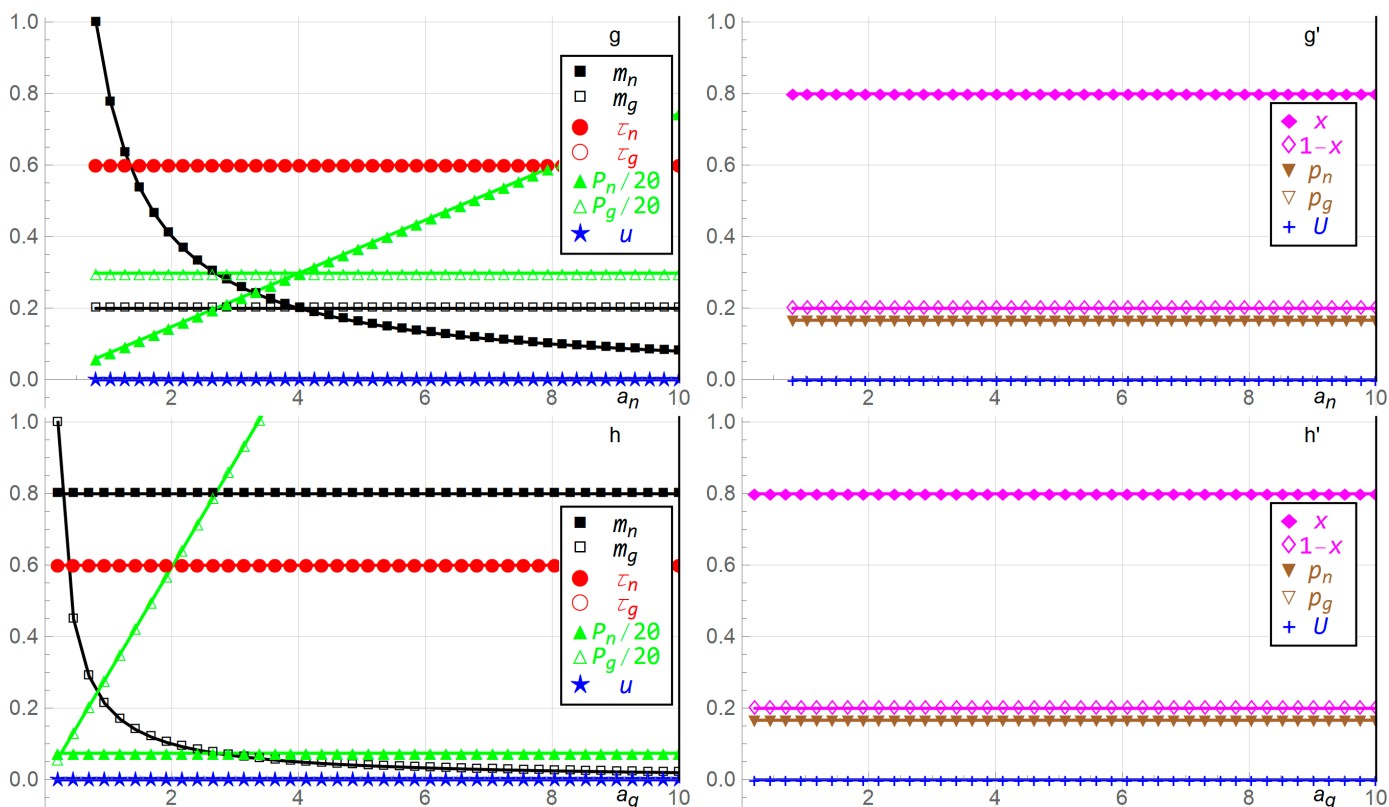

**Figure 4.** The government's monitoring probability $m_j$, taxation $\tau_j$, penalty factor $P_j$, and expected utility $u$, and the household's fractions $x$ and $1 - x$ of currencies $n$ and $g$, probability $p_j$ of tax evasion on currency $j$, and expected utility $U$, as functions of the eight parameter values $\alpha, \lambda_n, \lambda_g, \beta_n, \gamma_n, \beta_g, a_n, a_g$ relative to the benchmark parameter values $\alpha = 4/5$, $\lambda_n = \lambda_g = 1/5$, $\beta_n = \gamma_n = 2/5$, $\beta_g = 1/10$, $a_n = a_g = 1$. The eight double panels, for the eight parameters $\alpha, \lambda_n, \lambda_g, \beta_n, \gamma_n, \beta_g, a_n, a_g$, are referred to as (**a**,**a′**); (**b**,**b′**); (**c**,**c′**); (**d**,**d′**); (**e**,**e′**); (**f**,**f′**); (**g**,**g′**); and (**h**,**h′**). Division of $P_j$ with 20 is for scaling purposes, $j = n, g$.

In Figure 4a,a′, as the household's output elasticity $\alpha$ for currency $n$ increases, the household's fraction $x$ of currency $n$ increases linearly, the government's monitoring probability $m_n$ of currency $n$ increases linearly, and the government's penalty factor $P_g$ imposed on each household's holding of currency $g$ increases convexly; while the household's fraction $1 - x$ of currency $g$ decreases linearly, the government's monitoring probability $m_g$ of currency $g$ decreases linearly, and the government's penalty factor $P_n$ imposed on each household's holding of currency $n$ decreases convexly; and the remaining variables are constant.

In Figure 4b,b′,c,c′, as the exponential tax evasion parameter $\lambda_j$ increases, the household's probability $p_j$ of tax evasion on currency $j$ increases concavely, and the government's taxation $\tau_j$ on currency $j$ increases linearly; the government's penalty factor $P_j$ imposed on each household's holding of currency $j$ is relatively constant, and the remaining variables are constant, $j = n, g$.

In Figure 4d,d′, as the government's output elasticity $\beta_n$ for currency $n$ increases, the household's probabilities $p_n$ and $p_g$ of tax evasion on currencies $n$ and $g$ increase linearly and convexly, and the government's taxation $\tau_n$ and $\tau_g$ on currencies $n$ and $g$ decrease concavely and convexly. That causes taxation $\tau_g$ to be quite low when $\beta_n$ is high, since the government then taxes currency $n$ more than currency $g$. Furthermore, increasing $\beta_n$ causes the government's penalty factors $P_n$ and $P_g$ imposed on each household's holding of currencies $n$ and $g$ to increase concavely and convexly, and the remaining variables are constant.

In Figure 4e,e′, as the government's output elasticity $\beta_g$ for currency $g$ increases, the household's probability $p_g$ of tax evasion on currency $g$ increases convexly, the govern-

ment's taxation $\tau_g$ on currency $g$ decreases linearly, and the government's penalty factor $P_g$ imposed on each household's holding of currency $g$ increases convexly. The remaining variables are constant.

In Figure 4f,f', as the government's output elasticity $\gamma_n$ for currency $n$ when valuing taxation $\tau_n$ and valuing penalty $P_n$ on unsuccessful tax evasion on currency $n$ increases, the household's probabilities $p_n$ and $p_g$ of tax evasion on currencies $n$ and $g$ decreases concavely and increases convexly. Furthermore, as $\gamma_n$ increases, the government's taxation $\tau_n$ and $\tau_g$ on currencies $n$ and $g$ increases concavely and decreases convexly, the government's penalty factors $P_n$ and $P_g$ imposed on each household's holding of currencies $n$ and $g$ decreases concavely and increases convexly, and the remaining variables are constant.

In Figure 4g,g',4h,h', as the government's unit cost $a_j$ of choosing the monitoring probability $m_j$ of currency $j$ increases, the government's monitoring probability $m_j$ of currency $j$ decreases concavely, the government's penalty factor $P_j$ imposed on each household's holding of currency $j$ increases linearly, and the remaining variables are constant, $j = n, g$.

## 5. Discussion, Economic Intuition, and Policy Implications

Eight results in the previous section are particularly noteworthy. First, the household's fraction $x$ of the national currency $n$, the government's monitoring probability $m_n$ of the national currency $n$, and the penalty factor $P_g$ imposed on holding the global currency $g$, increase linearly, linearly, and convexly in the household's output elasticity $\alpha$ for the national currency $n$. It is assumed that as one currency becomes more important, valuable, and useful for the household, it holds more of it, which causes the government to monitor it more thoroughly. More extensive monitoring of one currency is accompanied with a lower penalty factor for that currency, and a higher penalty factor for the other currency. This inverse correlation between monitoring $m_j$ and the penalty factor $P_j$, shown in mboxfigfig:games-1159039-f004a,a', causes the household to choose a constant probability $p_j$ of tax evasion on currency $j$. The policy implication is that governments should be cognizant of this inverse correlation between monitoring $m_j$ and the penalty factor $P_j$, which can be implemented in laws and procedures. For example, increased monitoring $m_j$ without decreasing the penalty factor $P_j$ as shown in Figure 4a,a' cannot be expected to cause the household to choose a constant probability $p_j$ of tax evasion on currency $j$, but can instead cause the household to choose a lower probability $p_j$ of tax evasion on currency $j$ since the penalty factor $P_j$ is too high.

Second and third, the household's probability $p_j$ of tax evasion and the government's taxation $\tau_j$ increase concavely and linearly, respectively, in the exponential tax evasion parameter $\lambda_j$ for each currency $j$. The mathematical reason can be seen from Equation (2) where higher $\lambda_j$ causes lower $p_j^{\lambda_j}$, since $0 \leq p_j \leq 1$, which dilutes the impact of monitoring $m_j$ and the penalty factor $P_j$ through the term $m_j \tau_j P_j p_j^{\lambda_j}$, causing higher probability $p_j$ of tax evasion. The government's natural response is to tax more, which is expressed with higher $\tau_j$. The intuition is that if the government's structure of monitoring and penalties becomes more lenient, expressed with higher $\lambda_j$, the household will evade tax more, and will face higher taxation. The policy implication is that governments should holistically recognize the relationship between monitoring, penalties, the amount of taxation, and how households evade tax under these conditions.

Fourth, the household's probabilities $p_n$ and $p_g$ of tax evasion on both currencies $n$ and $g$ increase in the government's output elasticity $\beta_n$ for the national currency $n$. Furthermore, the government's taxation $\tau_j$ on both currencies decrease, and the penalty factor $P_j$ increase, in $\beta_n$. Additionally, a high $\beta_n$ eventually induces the government to tax that currency $n$ more than the global currency $g$. Since higher $\beta_n$ means that the government identifies more with the household, and thus becomes more altruistic, it is assumed that the household exploits the government's altruism through more tax evasion. Additionally, the household enjoys less taxation, although the government eventually taxes currency $n$, which it values, more than currency $g$, and eventually suffers higher penalties. The policy implication is

that governments should realize that identifying too much with households, by becoming more altruistic, and lowering taxes, with a possible objective of appeasing citizens and ensuring reelection, might cause the households to exploit the situation by evading tax even more.

Fifth, and similarly fourth, the household's probability $p_g$ of tax evasion on currency $g$ increases in the government's output elasticity $\beta_g$. The government's taxation $\tau_g$ on currency $g$ decreases in $\beta_g$, as the government identifies more strongly with the household. The government's penalty factor $P_g$ imposed on each household's holding of currency $g$ increases in $\beta_g$. The intuition is again that the household exploits the government's altruism through more tax evasion, enjoys less taxation, although eventually there is more taxation on the currency that the government values most, and eventually suffers higher penalties. The policy implication is again that governments should recognize the relationship between being altruistic, being exploited through different probabilities of tax evasion on the two currencies, and imposing adequate taxes and penalties.

Sixth, the household's probabilities $p_n$ and $p_g$ of tax evasion on currencies $n$ and $g$ decreases and increases in the government's output elasticity $\gamma_n$ for currency $n$, which values taxation $\tau_n$ and penalty $P_n$ on unsuccessful tax evasion on currency $n$. Furthermore, the household is less likely to evade tax on the national currency $n$ if the government values taxation $\tau_n$ and penalty $P_n$, expressed with $\gamma_n$, on the national currency $n$. The results are opposite for currency $g$, as shown in Sections 3 and 4. The intuition is that a higher $\gamma_n$, which implies valuing taxation and penalties for tax evasion, causes the government to be less altruistic towards the household regarding the national currency $n$, which causes more taxation with a lower associated penalty factor, and less tax evasion. Intuitively, higher $\gamma_n$ has the opposite impact for the global currency $g$. The policy implication is that governments should assess how they value taxation and penalties for tax evasion, which impacts how households evade tax differently on national and global currencies.

Seventh and eighth, the government's monitoring probability $m_j$ of each currency $j$ decreases in the unit cost $a_j$ of monitoring, counteracted by the penalty factor $P_j$ imposed on each household's holding of each currency increase. This causes the tax rates $\tau_n$ and $\tau_g$ and the household's probabilities $p_n$ and $p_g$ of tax evasion to be constant. The intuition is that the government compensates for a low (high) monitoring probability $m_j$, as regulated by the unit cost $a_j$ of monitoring, by choosing a high (low) penalty factor $P_j$. The model thus predicts, for example, that if the government is less able to monitor transactions and enforce regulations in cryptocurrencies, expressed by a high unit costs of monitoring, then it should impose higher penalties on each household's holding of cryptocurrencies when taxes are evaded. Whether that happens in practice is an interesting empirical question that should be analyzed in future research. For example, if the government's unit cost $a_g$ of monitoring in Figure 4g,g' is extremely high causing the monitoring probability $m_g$ to be extremely low, then a variety of consequences are possible. For example, the government might not be able to impose and enforce payment of sufficiently high penalties as predicted by the model, due to laws, regulations, and customs placing upper bounds on penalties, or households being unable to pay excessive penalties, for example. Alternatively, households might in practice not follow the expected utility theory when facing an extremely low monitoring probability $m_g$ of being detected and prosecuted for tax evasion, and might choose to ignore the probability of being monitored. The policy implication is that governments should be cognizant of the relationship between how they choose monitoring efforts and penalties for tax evasion, and how this relationship impacts their own taxation and the households' tax evasion.

## 6. Conclusions

This article presents a game between a government and a representative household holding two currencies, which can generally be any two assets, subject to taxation. The two currencies are a national currency, e.g., a CBDC and a global currency, e.g., Bitcoin, Zcash, or Facebook's Diem, which might have limited usage within a nation. The global currency

might offer other opportunities, e.g., tax evasion, user autonomy, discretion, peer-to-peer focus, no banking fees, payment on the black market, criminal activities, and potential return.

The household makes three strategic choices to maximize its Cobb Douglas expected utility with two output elasticities associated with the two currencies. Due to the different opportunities, usage, values, etc. provided by the two currencies, the household chooses to hold one fraction in the national currency, and the remaining fraction in the global currency. Additionally, the household chooses the tax evasion probability on each currency.

The government makes six strategic choices, i.e., the probability of detecting and prosecuting tax evasion on each currency, the tax rate on each currency, and the penalty factor imposed on each household when tax evasion is successfully detected and prosecuted for each currency. The government has a Cobb Douglas expected utility with four output elasticities, minus costs of choosing the monitoring probabilities. Two output elasticities are associated with the two currencies as the government identifies with the household. The two remaining output elasticities are due to the government benefitting from taxes and penalties. The government incurs a cost of choosing the monitoring probability.

The article analytically determines the players' nine strategic choices and expected utilities. Many results are in line with logic. Some results illustrate aspects that the governments and households should be cognizant of. The household prefers low taxation. The government identifies partly with each household, since it is either elected by the households or needs support from the households, but also needs income from taxation and might receive penalty payments for detecting tax evasion. The players' strategic choices are closely related to their output elasticities for the two currencies, and to the government's output elasticities that value taxation and penalties for tax evasion.

The household's fraction of the national currency, the government's monitoring probability of the national currency, and the penalty factor imposed on the global currency, increase the household's output elasticity for the national currency. The household's probability of tax evasion and the government's taxation increase in the exponential tax evasion parameter for each currency. The household's probabilities of tax evasion on both currencies increase in the government's output elasticity for the national currency. The government's taxation on both currencies decrease in the output elasticity for the national currency.

High output elasticity for the national currency eventually induces the government to tax that currency more than the global currency. The household's probability of tax evasion on the global currency increases in the government's output elasticity for that currency. The household is less (more) likely to tax evade on the national (global) currency if the government values taxation and penalty on the national (global) currency. The government's monitoring probability of each currency decreases in the unit cost of monitoring. The government's penalty factor imposed on each household's holding of each currency increases in the unit cost of monitoring. The results are illustrated numerically where each of eight parameter values are varied relative to a benchmark.

Future research should compile and assess empirical support for how households and governments choose strategies for national and global currencies, and assess common output elasticities in Cobb Douglas expected utilities for currencies. Such empirical support should be assessed against the fractions that a representative household chooses for each currency, and the probabilities the households choose for tax evasion on currencies. The government's probability of detecting and prosecuting tax evasion, the tax rate, and the penalty factor imposed on each household when tax evasion is successfully detected and prosecuted, should be empirically assessed for each currency.

Future research might also model more than two currencies, and additional players such as firms, multiple governments in multiple countries, central banks, banks, and international financial institutions. Various alternatives to the players' expected utilities might be evaluated, i.e., backing, convenience, confidentiality, transaction efficiency, financial stability, and security, as perceived by each player. More complexity and multiple time periods might also be incorporated.

**Author Contributions:** Conceptualization, G.W. and K.H.; methodology, G.W. and K.H.; software, K.H.; validation, K.H.; formal analysis, G.W. and K.H.; investigation, G.W. and K.H.; resources, K.H.; data curation, K.H.; writing—original draft preparation, G.W. and K.H.; writing—review and editing, K.H.; visualization, G.W. and K.H.; supervision, K.H.; project administration, K.H. All authors have read and agreed to the published version of the manuscript.

**Funding:** This research received no external funding.

**Institutional Review Board Statement:** Not applicable.

**Informed Consent Statement:** Not applicable.

**Data Availability Statement:** All data generated or analyzed during this study is included in this published article.

**Conflicts of Interest:** The authors declare no conflict of interest.

## Appendix A. Nomenclature

*Appendix A.1. Parameters*

$j$   Currency of kind $j$, $j = n, g$.
$n$   National currency.
$g$   Global currency.
$\alpha$   The household's output elasticity for currency $n$, $0 \leq \alpha \leq 1$.
$1 - \alpha$   The household's output elasticity for currency $g$, $0 \leq \alpha \leq 1$.
$\lambda_j$   Exponential tax evasion parameter, $0 \leq \lambda_j \leq \infty$, $j = n, g$.
$\beta_n$   The government's output elasticity for currency $n$ when identifying with the household, $0 \leq \beta_n \leq 1$.
$\beta_g$   The government's output elasticity for currency $g$ when identifying with the household, $0 \leq \alpha \leq 1$.
$\gamma_n$   The government's output elasticity for currency $n$ when valuing taxation $\tau_n$ and valuing penalty $P_n$ on unsuccessful tax evasion on currency $n$, $0 \leq \gamma_n \leq 1$.
$1 - \beta_n - \beta_g - \gamma_n$   The government's output elasticity for currency $g$ when valuing taxation and valuing penalties on unsuccessful tax evasion, of currency $g$, $0 \leq 1 - \beta_n - \beta_g - \gamma_n \leq 1$.
$a_j$   Unit cost of choosing the monitoring probability $n_j$, $a_j \geq 0$, $j = n, g$.

*Appendix A.2. Household's Free Choice Variables*

$p_j$   Household's probability of tax evasion on currency $j$, $j = n, g$, $0 \leq p_j \leq 1$.
$x$   Household's fraction of currency $n$, $0 \leq x \leq 1$.

*Appendix A.3. Government's Free Choice Variables*

$m_j$   Government's probability of monitoring and thus detecting and prosecuting tax evasion on currency $j$, $0 \leq m_j \leq 1$, $j = n, g$.
$\tau_j$   Household's tax rate on currency $j$, $0 \leq \tau_j \leq 1$, $j = n, g$.
$P_j$   Government's penalty factor imposed on each household's holding of currency $j$ when tax evasion is successfully detected and prosecuted, $j = n, g$.

*Appendix A.4. Dependent Variables*

$U$   Household's expected utility.
$u$   Government's expected utility per household.
$1 - x$   Household's fraction of currency $g$, $0 \leq x \leq 1$.

**Appendix B. Determining the Household's Free Choice Variables**

Differentiating the household's expected utility $U$ in Equation (2) with respect to its free choice variable $x$ gives

$$\frac{\partial U}{\partial x} = \begin{cases} \left(1 - (1 - p_n)\tau_n - m_n\tau_n P_n p_n^{\lambda_n}\right)^\alpha \left(1 - (1 - p_g)\tau_g - m_g\tau_g P_g p_g^{\lambda_g}\right)^{1-\alpha} \\ \qquad \times \frac{(\alpha - x)x^{\alpha-1}}{(1-x)^\alpha} \ if \ P_j \leq \frac{1 - (1 - p_j)\tau_j}{m_j\tau_j p_j^{\lambda_j}}, j = n, g \\ 0 \ otherwise \end{cases} \tag{A1}$$

which is equated with zero and solved to yield Equation (5). The second order conditions, inserting $x = \alpha$, are satisfied as negative, i.e.,

$$\frac{\partial^2 U}{\partial x^2}\bigg|_{x=\alpha} = \begin{cases} -\left(1 - (1 - p_n)\tau_n - m_n\tau_n P_n p_n^{\lambda_n}\right)^\alpha \left(1 - (1 - p_g)\tau_g - m_g\tau_g P_g p_g^{\lambda_g}\right)^{1-\alpha} \\ \qquad \times \frac{\alpha^{\alpha-1}}{(1-\alpha)^\alpha} \ if \ P_j \leq \frac{1 - (1 - p_j)\tau_j}{m_j\tau_j p_j^{\lambda_j}}, j = n, g \\ 0 \ otherwise \end{cases} \tag{A2}$$

Differentiating the household's expected utility $U$ in Equation (2) with respect to its free choice variables $p_j$, $j = n, g$, gives

$$\frac{\partial U}{\partial p_n} = \begin{cases} \alpha\tau_n\left(1 - m_n P_n \lambda_n p_n^{\lambda_n-1}\right)\left(1 - (1 - p_n)\tau_n - m_n\tau_n P_n p_n^{\lambda_n}\right)^{\alpha-1} \\ \quad \times \left(1 - (1 - p_g)\tau_g - m_g\tau_g P_g p_g^{\lambda_g}\right)^{1-\alpha} x^\alpha (1-x)^{1-\alpha} \\ \qquad if \ P_j \leq \frac{1 - (1 - p_j)\tau_j}{m_j\tau_j p_j^{\lambda_j}}, j = n, g \\ 0 \ otherwise \end{cases} \tag{A3}$$

and

$$\frac{\partial U}{\partial p_g} = \begin{cases} (1 - \alpha)\tau_g\left(1 - m_g P_g \lambda_g p_g^{\lambda_g-1}\right)\left(1 - (1 - p_n)\tau_n - m_n\tau_n P_n p_n^{\lambda_n}\right)^\alpha \\ \quad \times \left(1 - (1 - p_g)\tau_g - m_g\tau_g P_g p_g^{\lambda_g}\right)^{-\alpha} x^\alpha (1-x)^{1-\alpha} \\ \qquad if \ P_j \leq \frac{1 - (1 - p_j)\tau_j}{m_j\tau_j p_j^{\lambda_j}}, j = n, g \\ 0 \ otherwise \end{cases} \tag{A4}$$

which are equated with zero and solved to yield Equation (6). The second order condition for $p_n$ is satisfied as negative, i.e.,

$$\frac{\partial^2 U}{\partial p_n^2} = \begin{cases} -\left(1 - (1 - p_n)\tau_n - m_n\tau_n P_n p_n^{\lambda_n}\right)^{\alpha-2}\left(1 - (1 - p_g)\tau_g - m_g\tau_g P_g p_g^{\lambda_g}\right)^{1-\alpha} \\ \times \alpha\tau_n(1-x)^{1-\alpha} x^\alpha \left(m_n P_n p_n^{\lambda_n-2}(\lambda_n - 1)\lambda_n\left(1 - (1 - p_n)\tau_n - m_n\tau_n P_n p_n^{\lambda_n}\right)\right. \\ \left. \quad + (1 - \alpha)\tau_n\left(1 - m_n P_n \lambda_n p_n^{\lambda_n-1}\right)^2\right) \ if \ P_n \leq \frac{1 - (1 - p_n)\tau_n}{m_n\tau_n p_n^{\lambda_n}} \\ 0 \ otherwise \end{cases} \tag{A5}$$

The second order condition for $p_g$ is analogous.

## Appendix C. Determining the Government's Free Choice Variables

Differentiating the government's expected utility $u$ in Equation (3) with respect to its six free choice variables $m_j, \tau_j, P_j, j = n, g$, gives

$$N \equiv \left(1 - p_n + m_n P_n p_n^{\lambda_n}\right) \tau_n x, \quad G \equiv \left(1 - p_g - m_g P_g p_g^{\lambda_g}\right) \tau_g (1 - x),$$

$$\frac{\partial u}{\partial m_n} = \begin{cases} -x(1 - x - G)^{\beta_g} (G - a_g m_g)^{1 - \beta_n - \beta_g - \gamma_n} (x - N)^{\beta_n - 1} (N - a_n m_n)^{\gamma_n - 1} \\ \times \begin{pmatrix} m_n p_n^{2\lambda_n} P_n^2 x (\beta_n + \gamma_n) \tau_n^2 + a_n \gamma_n (1 + (-1 + p_n) \tau_n) \\ -p_n^{\lambda_n} P_n \tau_n (a_n m_n (\beta_n + \gamma_n) + x(\gamma_n + (-1 + p_n)(\beta_n + \gamma_n) \tau_n)) \end{pmatrix} \\ \qquad \qquad if \ P_j \leq \frac{1 - (1 - p_j) \tau_j}{m_j \tau_j p_j^{\lambda_j}}, j = n, g \\ 0 \ otherwise \end{cases} \qquad (A6)$$

and

$$\frac{\partial u}{\partial m_g} = \begin{cases} (-1 + x)(1 - x - G)^{\beta_g - 1} (G - a_g m_g)^{-\beta_n - \beta_g - \gamma_n} (x - N)^{\beta_n} (N - a_n m_n)^{\gamma_n} \\ \times (a_g m_g p_g^{\lambda_g} P_g (-1 + \beta_n + \gamma_n) \tau_g - a_g (-1 + \beta_g + \beta_n + \gamma_n) \\ \times (1 + (-1 + p_g) \tau_g) + p_g^{\lambda_g} P_g (-1 + x) \tau_g (-\beta_g \\ +(-1 + \beta_n + \gamma_n)(-1 + \tau_g + p_g (-1 + m_g p_g^{\lambda_g - 1} P_g) \tau_g))) \\ \qquad \qquad if \ P_j \leq \frac{1 - (1 - p_j) \tau_j}{m_j \tau_j p_j^{\lambda_j}}, j = n, g \\ 0 \ otherwise \end{cases} \qquad (A7)$$

and

$$\frac{\partial u}{\partial \tau_n} = \begin{cases} -(1 + p_n (-1 + m_n p_n^{\lambda_n - 1} P_n)) x(1 - x - G)^{\beta_g} (x - N)^{\beta_n - 1} \\ \times (G - a_g m_g)^{1 - \beta_n - \beta_g - \gamma_n} (N - a_n m_n)^{\gamma_n - 1} (-a_n m_n \beta_n - x \gamma_n \\ +(1 + p_n (-1 + m_n p_n^{\lambda_n - 1} P_n)) x(\beta_n + \gamma_n) \tau_n)) \\ \qquad \qquad if \ P_j \leq \frac{1 - (1 - p_j) \tau_j}{m_j \tau_j p_j^{\lambda_j}}, j = n, g \\ 0 \ otherwise \end{cases} \qquad (A8)$$

and

$$\frac{\partial u}{\partial \tau_g} = \begin{cases} \left(1 + p_g (-1 + m_g p_g^{\lambda_g - 1} P_g)\right)(-1 + x)(1 - x - G)^{\beta_g - 1} (G - a_g m_g)^{-\beta_n - \beta_g - \gamma_n} \\ \times (\beta_g - a_g m_g \beta_g - x \beta_g - (-1 + \beta_n + \gamma_n)(-1 + \tau_g + p_g(-1 + \\ m_g p_g^{\lambda_g - 1} P_g) \tau_g) + x(-1 + \beta_n + \gamma_n)(-1 + \tau_g \\ +p_g (-1 + m_g p_g^{\lambda_g - 1} P_g) \tau_g))(x - N)^{\beta_n} (N - a_n m_n)^{\gamma_n} \\ \qquad \qquad if \ P_j \leq \frac{1 - (1 - p_j) \tau_j}{m_j \tau_j p_j^{\lambda_j}}, j = n, g \\ 0 \ otherwise \end{cases}$$
$$(A9)$$

and

$$\frac{\partial u}{\partial P_n} = \begin{cases} -(1 - x - G)^{\beta_g} (G - a_g m_g)^{1 - \beta_n - \beta_g - \gamma_n} (x - N)^{\beta_n - 1} (N - a_n m_n)^{\gamma_n - 1} \\ \times m_n p_n^{\lambda_n} x \tau_n \left(-a_n m_n \beta_n - x \gamma_n + \left(1 + p_n (-1 + m_n p_n^{\lambda_n - 1} P_n)\right) x(\beta_n + \gamma_n) \tau_n\right) \\ \qquad \qquad if \ P_j \leq \frac{1 - (1 - p_j) \tau_j}{m_j \tau_j p_j^{\lambda_j}}, j = n, g \\ 0 \ otherwise \end{cases} \qquad (A10)$$

and

$$
\frac{\partial u}{\partial P_g} = \begin{cases}
m_g p_g^{\lambda_g}(-1+x)\tau_g(1-x-G)^{\beta_g-1}(G-a_g m_g)^{-\beta_n-\beta_g-\gamma_n}(\beta_g - a_g m_g \beta_g \\
-x\beta_g - (-1+\beta_n+\gamma_n)\left(-1+\tau_g+p_g\left(-1+m_g p_g^{\lambda_g-1}P_g\right)\tau_g\right) + x(-1+\beta_n+ \\
\gamma_n)(-1+\tau_g+p_g(-1+m_g p_g^{\lambda_g-1}P_g)\tau_g))(x-N)^{\beta_n}(N-a_n m_n)^{\gamma_n} \\
\qquad\qquad\qquad if\ P_j \le \dfrac{1-(1-p_j)\tau_j}{m_j \tau_j p_j^{\lambda_j}}, j=n,g \\
0\ otherwise
\end{cases}
\tag{A11}
$$

Equating the first order conditions in Equations (A6)–(A11) with zero and solving gives Equation (7), which are valid when the inequalities are satisfied. The if-test $P_j \le \dfrac{1-(1-p_j)\tau_j}{m_j \tau_j p_j^{\lambda_j}}, j=n,g$, is omitted in Equation (7) since it is always satisfied. It can be shown that the second order conditions are satisfied as negative.

## Appendix D. Proof of Property 1

Equations (5)–(7) constitute nine equations with the nine unknown variables $x$, $p_j$, $m_j$, $\tau_j$, $P_j$, $j=n,g$, which are solved to yield Equation (8). Just as the if-test $P_j \le \dfrac{1-(1-p_j)\tau_j}{m_j \tau_j p_j^{\lambda_j}}, j = n,g$, is omitted in Equation (7) since it is always satisfied, it is also omitted for $x$, $p_n$ and $p_g$ in Equation (8) since it is always satisfied. The inequalities $a_n \ge \alpha$ and $a_g \ge 1-\alpha$ follow from Equation (7) when $x = \alpha$. The inequality $0 \le \lambda_j \le 1$, $j=n,g$, follows since $\lambda_j > 1$ would cause taxation $\tau_n > 1$ in Equation (8), which is not meaningful.

## Appendix E. First Order and Second Order Derivatives for Property 2

Differentiating Equation (8) when $a_n \ge \alpha$, $a_g \ge 1-\alpha$, $0 \le \lambda_j \le 1$, $j=n,g$, gives

$$
\begin{aligned}
&\frac{\partial x}{\partial \alpha} = 1,\ \frac{\partial(1-x)}{\partial \alpha} = -1,\ \frac{\partial p_n}{\partial \alpha} = \frac{\partial p_g}{\partial \alpha} = \frac{\partial \tau_n}{\partial \alpha} = \frac{\partial \tau_g}{\partial \alpha} = 0,\ \frac{\partial m_n}{\partial \alpha} = \frac{1}{a_n},\ \frac{\partial m_g}{\partial \alpha} = \frac{-1}{a_g}, \\
&\frac{\partial P_n}{\partial \alpha} = \frac{-a_n}{\lambda_n \alpha^2}\left(\frac{\lambda_n \beta_n}{\lambda_n \beta_n + \gamma_n}\right)^{1-\lambda_n},\ \frac{\partial^2 P_n}{\partial \alpha^2} = \frac{2a_n}{\lambda_n \alpha^3}\left(\frac{\lambda_n \beta_n}{\lambda_n \beta_n + \gamma_n}\right)^{1-\lambda_n}, \\
&\frac{\partial P_g}{\partial \alpha} = \frac{a_g}{\lambda_g(1-\alpha)^2}\left(\frac{\lambda_g \beta_g}{1-\beta_n-\gamma_n-(1-\lambda_g)\beta_g}\right)^{1-\lambda_g}, \\
&\frac{\partial^2 P_g}{\partial \alpha^2} = \frac{2a_g}{\lambda_g(1-\alpha)^3}\left(\frac{\lambda_g \beta_g}{1-\beta_n-\gamma_n-(1-\lambda_g)\beta_g}\right)^{1-\lambda_g}
\end{aligned}
\tag{A12}
$$

$$
\begin{aligned}
&\frac{\partial x}{\partial \lambda_n} = \frac{\partial(1-x)}{\partial \lambda_n} = \frac{\partial p_g}{\partial \lambda_n} = \frac{\partial m_n}{\partial \lambda_n} = \frac{\partial m_g}{\partial \lambda_n} = \frac{\partial \tau_g}{\partial \lambda_n} = \frac{\partial P_g}{\partial \lambda_n} = 0, \\
&\frac{\partial p_n}{\partial \lambda_n} = \frac{\beta_n \gamma_n}{(\lambda_n \beta_n + \gamma_n)^2},\ \frac{\partial^2 p_n}{\partial \lambda_n^2} = \frac{-2\beta_n^2 \gamma_n}{(\lambda_n \beta_n + \gamma_n)^3},\ \frac{\partial \tau_n}{\partial \lambda_n} = \frac{\beta_n}{\beta_n + \gamma_n}, \\
&\frac{\partial P_n}{\partial \lambda_n} = \frac{-a_n \beta_n}{\alpha(\lambda_n \beta_n + \gamma_n)^2}\left(\frac{\lambda_n \beta_n}{\lambda_n \beta_n + \gamma_n}\right)^{-\lambda_n}\left(\beta_n + \gamma_n + (\lambda_n \beta_n + \gamma_n)Ln\left(\frac{\lambda_n \beta_n}{\lambda_n \beta_n + \gamma_n}\right)\right)
\end{aligned}
\tag{A13}
$$

$$
\begin{aligned}
&\frac{\partial x}{\partial \lambda_g} = \frac{\partial(1-x)}{\partial \lambda_g} = \frac{\partial p_n}{\partial \lambda_g} = \frac{\partial m_n}{\partial \lambda_g} = \frac{\partial m_g}{\partial \lambda_g} = \frac{\partial \tau_n}{\partial \lambda_g} = \frac{\partial P_n}{\partial \lambda_g} = 0, \\
&\frac{\partial p_g}{\partial \lambda_g} = \frac{\beta_g(1-\beta_n-\beta_g-\gamma_n)}{(1-\beta_n-\gamma_n-(1-\lambda_g)\beta_g)^2},\ \frac{\partial^2 p_g}{\partial \beta_g^2} = \frac{2\beta_g^2(1-\beta_n-\beta_g-\gamma_n)}{(1-\beta_n-\gamma_n-(1-\lambda_g)\beta_g)^3}, \\
&\frac{\partial \tau_g}{\partial \lambda_g} = \frac{\beta_g}{1-\beta_n-\gamma_n}, \\
&\frac{\partial P_g}{\partial \lambda_g} = \frac{a_g \beta_g}{(1-\alpha)(1-\beta_n-\gamma_n-(1-\lambda_g)\beta_g)^2}\left(\frac{\lambda_g \beta_g}{1-\beta_n-\gamma_n-(1-\lambda_g)\beta_g}\right)^{-\lambda_g} \\
&\qquad\times\left(1-\beta_n-\gamma_n+(1-\beta_n-\gamma_n-(1-\lambda_g)\beta_g)Ln\left(\frac{\lambda_g \beta_g}{1-\beta_n-\gamma_n-(1-\lambda_g)\beta_g}\right)\right)
\end{aligned}
\tag{A14}
$$

$$\frac{\partial x}{\partial \beta_n} = \frac{\partial(1-x)}{\partial \beta_n} = \frac{\partial m_n}{\partial \beta_n} = \frac{\partial m_g}{\partial \beta_n} = 0, \frac{\partial p_n}{\partial \beta_n} = \frac{\lambda_n \gamma_n}{(\lambda_n \beta_n + \gamma_n)^2}, \frac{\partial^2 p_n}{\partial \beta_n^2} = \frac{-2\lambda_n^2 \gamma_n}{(\lambda_n \beta_n + \gamma_n)^3},$$

$$\frac{\partial p_g}{\partial \beta_n} = \frac{\lambda_g \beta_g}{\left(1-\beta_n-\gamma_n-(1-\lambda_g)\beta_g\right)^2}, \frac{\partial^2 p_g}{\partial \beta_n^2} = \frac{2\lambda_g \beta_g}{\left(1-\beta_n-\gamma_n-(1-\lambda_g)\beta_g\right)^3},$$

$$\frac{\partial \tau_n}{\partial \beta_n} = \frac{-(1-\lambda_n)\gamma_n}{(\beta_n+\gamma_n)^2}, \frac{\partial^2 \tau_n}{\partial \beta_n^2} = \frac{2\gamma_n(1-\lambda_n)}{(\beta_n+\gamma_n)^3}, \frac{\partial \tau_g}{\partial \beta_n} = \frac{-(1-\lambda_g)\beta_g}{(1-\beta_n-\gamma_n)^2},$$

$$\frac{\partial^2 \tau_g}{\partial \beta_n^2} = \frac{-2\beta_g(1-\lambda_g)}{(1-\beta_n-\gamma_n)^3}, \frac{\partial P_n}{\partial \beta_n} = \frac{a_n(1-\lambda_n)\gamma_n}{\alpha(\lambda_n \beta_n+\gamma_n)^2}\left(\frac{\lambda_n \beta_n}{\lambda_n \beta_n+\gamma_n}\right)^{-\lambda_n}, \qquad \text{(A15)}$$

$$\frac{\partial^2 P_n}{\partial \beta_n^2} = \frac{-a_n \gamma_n(2\beta_n+\gamma_n)(1-\lambda_n)\lambda_n}{\alpha \beta_n(\lambda_n \beta_n+\gamma_n)^3}\left(\frac{\lambda_n \beta_n}{\lambda_n \beta_n+\gamma_n}\right)^{-\lambda_n},$$

$$\frac{\partial P_g}{\partial \beta_n} = \frac{a_g(1-\lambda_g)\beta_g}{(1-\alpha)\left(1-\beta_n-\gamma_n-(1-\lambda_g)\beta_g\right)^2}\left(\frac{\lambda_g \beta_g}{1-\beta_n-\gamma_n-(1-\lambda_g)\beta_g}\right)^{-\lambda_g},$$

$$\frac{\partial^2 P_g}{\partial \beta_n^2} = \frac{a_g \beta_g(2-\lambda_g)(1-\lambda_g)}{(1-\alpha)\left(1-\beta_n-\gamma_n-(1-\lambda_g)\beta_g\right)^3}\left(\frac{\lambda_g \beta_g}{1-\beta_n-\gamma_n-(1-\lambda_g)\beta_g}\right)^{-\lambda_g}$$

$$\frac{\partial x}{\partial \beta_g} = \frac{\partial(1-x)}{\partial \beta_g} = \frac{\partial p_n}{\partial \beta_g} = \frac{\partial m_n}{\partial \beta_g} = \frac{\partial m_g}{\partial \beta_g} = \frac{\partial \tau_n}{\partial \beta_g} = \frac{\partial P_n}{\partial \beta_g} = 0,$$

$$\frac{\partial p_g}{\partial \beta_g} = \frac{\lambda_g(1-\beta_n-\gamma_n)}{\left(1-\beta_n-\gamma_n-(1-\lambda_g)\beta_g\right)^2}, \frac{\partial^2 p_g}{\partial \beta_g^2} = \frac{2(1-\beta_n-\gamma_n)(1-\lambda_g)\lambda_g}{\left(1-\beta_n-\gamma_n-(1-\lambda_g)\beta_g\right)^3}, \qquad \text{(A16)}$$

$$\frac{\partial \tau_g}{\partial \beta_g} = \frac{-(1-\lambda_g)}{1-\beta_n-\gamma_n},$$

$$\frac{\partial P_g}{\partial \beta_g} = \frac{a_g(1-\beta_n-\gamma_n)(1-\lambda_g)}{(1-\alpha)\left(1-\beta_n-\gamma_n-(1-\lambda_g)\beta_g\right)^2}\left(\frac{\lambda_g \beta_g}{1-\beta_n-\gamma_n-(1-\lambda_g)\beta_g}\right)^{-\lambda_g}$$

$$\frac{\partial x}{\partial \gamma_n} = \frac{\partial(1-x)}{\partial \gamma_n} = \frac{\partial m_n}{\partial \gamma_n} = \frac{\partial m_g}{\partial \gamma_n} = 0, \frac{\partial p_n}{\partial \gamma_n} = \frac{-\lambda_n \beta_n}{(\lambda_n \beta_n+\gamma_n)^2}, \frac{\partial^2 p_n}{\partial \gamma_n^2} = \frac{2\lambda_n \beta_n}{(\lambda_n \beta_n+\gamma_n)^3},$$

$$\frac{\partial p_g}{\partial \gamma_n} = \frac{\lambda_g \beta_g}{\left(1-\beta_n-\gamma_n-(1-\lambda_g)\beta_g\right)^2}, \frac{\partial^2 p_g}{\partial \gamma_n^2} = \frac{2\lambda_g \beta_g}{\left(1-\beta_n-\gamma_n-(1-\lambda_g)\beta_g\right)^3},$$

$$\frac{\partial \tau_n}{\partial \gamma_n} = \frac{(1-\lambda_n)\beta_n}{(\beta_n+\gamma_n)^2}, \frac{\partial^2 \tau_n}{\partial \gamma_n^2} = \frac{-2\beta_n(1-\lambda_n)}{(\beta_n+\gamma_n)^3}, \frac{\partial \tau_g}{\partial \gamma_n} = \frac{-(1-\lambda_g)\beta_g}{(1-\beta_n-\gamma_n)^2},$$

$$\frac{\partial^2 \tau_g}{\partial \gamma_n^2} = \frac{-2\beta_g(1-\lambda_g)}{(1-\beta_n-\gamma_n)^2}, \frac{\partial P_n}{\partial \gamma_n} = \frac{-a_n(1-\lambda_n)\beta_n}{\alpha(\lambda_n \beta_n+\gamma_n)^2}\left(\frac{\lambda_n \beta_n}{\lambda_n \beta_n+\gamma_n}\right)^{-\lambda_n}, \qquad \text{(A17)}$$

$$\frac{\partial^2 P_n}{\partial \gamma_n^2} = \frac{a_n \beta_n(2-\lambda_n)(1-\lambda_n)}{\alpha(\lambda_n \beta_n+\gamma_n)^3}\left(\frac{\lambda_n \beta_n}{\lambda_n \beta_n+\gamma_n}\right)^{-\lambda_n},$$

$$\frac{\partial P_g}{\partial \gamma_n} = \frac{a_g(1-\lambda_g)\beta_g}{(1-\alpha)\left(1-\beta_n-\gamma_n-(1-\lambda_g)\beta_g\right)^2}\left(\frac{\lambda_g \beta_g}{1-\beta_n-\gamma_n-(1-\lambda_g)\beta_g}\right)^{-\lambda_g},$$

$$\frac{\partial^2 P_g}{\partial \gamma_n^2} = \frac{a_g \beta_g(2-\lambda_g)(1-\lambda_g)}{(1-\alpha)\left(1-\beta_n-\gamma_n-(1-\lambda_g)\beta_g\right)^2}\left(\frac{\lambda_g \beta_g}{1-\beta_n-\gamma_n-(1-\lambda_g)\beta_g}\right)^{-\lambda_g}$$

$$\frac{\partial x}{\partial a_n} = \frac{\partial(1-x)}{\partial a_n} = \frac{\partial p_n}{\partial a_n} = \frac{\partial p_g}{\partial a_n} = \frac{\partial m_g}{\partial a_n} = \frac{\partial \tau_n}{\partial a_n} = \frac{\partial \tau_g}{\partial a_n} = \frac{\partial P_g}{\partial a_n} = 0,$$

$$\frac{\partial m_n}{\partial a_n} = \frac{-\alpha}{a_n^2}, \frac{\partial^2 m_n}{\partial a_n^2} = \frac{2\alpha}{a_n^3}, \frac{\partial P_n}{\partial a_n} = \frac{1}{\alpha \lambda_n}\left(\frac{\lambda_n \beta_n}{\lambda_n \beta_n+\gamma_n}\right)^{1-\lambda_n} \qquad \text{(A18)}$$

$$\frac{\partial x}{\partial a_g} = \frac{\partial(1-x)}{\partial a_g} = \frac{\partial p_n}{\partial a_g} = \frac{\partial p_g}{\partial a_g} = \frac{\partial m_n}{\partial a_g} = \frac{\partial \tau_n}{\partial a_g} = \frac{\partial \tau_g}{\partial a_g} = \frac{\partial P_n}{\partial a_g} = 0,$$

$$\frac{\partial m_g}{\partial a_g} = \frac{-(1-\alpha)}{a_g^2}, \frac{\partial^2 m_g}{\partial a_g^2} = \frac{2(1-\alpha)}{a_g^2}, \qquad \text{(A19)}$$

$$\frac{\partial P_g}{\partial a_g} = \frac{1}{(1-\alpha)\lambda_g}\left(\frac{\lambda_g \beta_g}{1-\beta_n-\gamma_n-(1-\lambda_g)\beta_g}\right)^{1-\lambda_g}$$

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
