# Peer review of "Governmental Taxation of Households Choosing between a National Currency and a Cryptocurrency"

_games, doi:10.3390/g12020034_

Round 1

Reviewer 1 Report

I will start with the bottom line. It is worth criticizing it as part of a broader academic debate rather than me killing it during the review process.

The research is well written and well supported analytically. Perhaps too well.

I confess following the math was a little challenging, but in the end I did not detect anything outrageous.

Here are my major comments:

1. The model is a game between the government and a representative household. I suspect an alternative set up in which one would model the government, a network of households, and a network of corporations would be more relevant, but would raise the level of complexity to a height where finding a decent solution would be nearly impossible. The main merit of the model presented here is its tractability. This is a perfect example of substitution bias: because we cannot solve a complex problem we substitute it with another one, easier to solve.

2. The author(s) do not account for other costs associated with holding the two currencies. Gaining access to a virtual currency is not as straightforward as holding a government-issued currency. Just to purchase bitcoins, etheruems, etc. one has to get an electronic wallet, connect to a marketplace, perhaps download a blockchain node etc, etc. something beyond the competence of a majority of households. I would argue we deal with a segmented market.

3. I personally question the wisdom of modeling a financial asset in the framework of supply and demand. I would argue the differences between a government-issued currency and a cryptocurrency are more significant than what the paper let us believe. Nevertheless, they both represent financial assets, and as such, they do not live in the supply and demand ecosystem. I realize I am going against the grain here – a vast majority of economists follow this gospel – and hence, I cannot hold this against the authors. I am just pointing out that both the price and the decision to hold the asset are driven by expected returns. Supply and demand are simply the outward manifestation of these expectations, not the main drivers. Not to mention the words “optimizing,” and “maximizing,” make me cringe every time I read them - in general, not only here. That’s not how real economic agents behave. But again, this is widely accepted among economists.

4. The conclusions make sense, and appeal to our economic intuition. But I question the usefulness of putting up a math virtuosity and dexterity show to conclude water flows downstream. I would expect households be more inclined to engage in tax evasion in the case of the “global” currency simply because the government is less able to enforce regulation and monitor transactions in cryptocurrencies – at least for now. This is an overriding consideration, notwithstanding the conclusions of the model.

Minor comments:

Very important: many of the equations and graphs presented in the text should go in an appendix at the end.

Reviewer 2 Report

The paper is well executed and innovative. I enjoyed reading it. I do not want to suggest unnecessary changes in the paper.

Reviewer 3 Report

The paper studies a simultaneous move – one shot game with two players, a household and a government. The household decides strategically the composition of the currency portfolio to hold and the level of tax evasion for each currency. The government decides tax rates, auditing effort and penalties in the event of tax evasion being detected.

The paper attempts to combine interesting issues, specifically currency portfolio decisions (including a digital global currency) and tax evasion decisions of households, and strategic taxation and auditing of governments.

Unfortunately, the paper fails to provide a proper description of the literature and to explain how the contribution of the paper relates to it. The results are reported analytically and numerically, but they are not accompanied by sufficient economics intuition. I report my concerns in more detail below.

  1. Section 1.1 mentions cryptocurrencies. It would help the reader if the authors could provide a very short description of how these new currencies work, advantages and disadvantages. In addition, it would be helpful if the authors could briefly explain how tax evasion may take place in currency markets and how digital/global currencies may offer an advantage. Finally, is there any empirical/experimental evidence of the extent of tax evasion in currency markets?
  2. Sections 1.2. and 1.3 feel rather disjoint. How does the model in the paper relate to previous contributions? How does the paper contribute to the literature and to what branch in particular? Specific to section 1.3, the paper mentions a few unpublished works and fails to mention seminar contributions in the field of the economic analysis of tax evasion. To mention a few:

Allingham, M. G. and Sandmo, A., (1972), "Income tax evasion: a theoretical analysis", \textit{Journal of Public Economics},1(3-4), 323-338.

Andreoni, J., Erard, B. and Feinstein, J., (1998), "Tax compliance", \textit{Journal of Economic Literature}, 36, pages 818-860.

Becker, G., (1968), "Crime and Punishment: An Economic Approach\textquotedblright, \textit{Journal of Political Economy}, 76, pp. 169-217.

Dhami, S., and Al Nowaihi, A., (2007), \textquotedblleft Why do people pay taxes? Prospect theory versus expected utility theory\textquotedblright, \textit{Journal of Economic Behavior and Organization}, 64, 171--192.

Kleven, H. J., Knudsen, M.B., Thustrup Kreiner, C., Pedersen, S. and Saez, E., (2011), "Unwilling or unable to cheat? Evidence from a tax audit experiment in Denmark", \textit{Econometrica}, 79, pages 651-692.

Luttmer, E.F.P. and Singhal, M., (2014), "Tax morale", \textit{Journal of Economic Perspectives}, 28, pages 143-168.

Myles, G.D. and Naylor, R.A., (1996), "A model of tax evasion with group conformity and social customs", European Journal of Political Economy, 12, 49--66.

Slemrod, J., Yitzhaki, S., (2002), "Tax avoidance, evasion, and administration". In A. J. Auerbach and M. Feldstein (Eds.), \textit{Handbook of Public Economics}, edition 1, volume 3, chapter 22, pages 1423-1470 Elsevier.

Torgler, B., (2002), "Speaking to Theorists and Searching for Facts: Tax

Morale and Tax Compliance in Experiments", \textit{Journal of Economic

Surveys}, 16, 657-684.

Yitzhaki, S., (1974), "Income tax evasion: a theoretical analysis", \textit{Journal of Public Economics}, 3(2), 201-202.

  1. Section 2 describes the model. The choice of the particular assumptions should be better motivated. In particular, the function forms of the utilities (why Cobb-Douglas? why both players?) and the altruistic nature of the government. I do not find it fully convincing to imagine a government that internalise the advantage for households of evading taxes.
  2. Section 3 and 4 produce and describe the equilibrium of the game and comparative statics. Leaving aside cosmetic issues (it is really not pleasant to read pages of cumbersome mathematical expressions followed by numerical examples), the main problem is that the results are not accompanied by any economic intuition, nor a discussion of the economic/policy implications. After reading these sections (and the whole paper), the reader cannot find clear answers to questions such as: under what circumstances tax evasion can be reduced? What policies can be more effective and under what circumstances? What institutional setting favours compliance? etc.

Round 2

Reviewer 3 Report

The authors have addressed all the points that I had raised in my previous report. I appreciate in particular the introduction of more detailed economic intuition in relation to the results of the analysis.

Minor typos: 

  1. Page 9, "Buying cryptocurrencies have become" should read "Buying cryptocurrencies has become"
  2. Page 16, section 5. "First, The household's" should read "First, the household's".

Author Response

We addressed all revisions as suggested.